# CITED: A Decision Boundary-Aware Signature for GNNs Towards Model Extraction Defense

## Abstract

Graph neural networks (GNNs) have demonstrated superior performance in various applications, such as recommendation systems and financial risk management. However, deploying large-scale GNN models locally is particularly challenging for users, as it requires significant computational resources and extensive property data. Consequently, Machine Learning as a Service (MLaaS) has become increasingly popular, offering a convenient way to deploy and access various models, including GNNs. However, an emerging threat known as Model Extraction Attacks (MEAs) presents significant risks, as adversaries can readily obtain surrogate GNN models exhibiting similar functionality. Specifically, attackers repeatedly query the target model using subgraph inputs to collect corresponding responses. These input-output pairs are subsequently utilized to train their own surrogate models at minimal cost. Many techniques have been proposed to defend against MEAs, but most are limited to specific output levels (e.g., embedding or label) and suffer from inherent technical drawbacks. To address these limitations, we propose a novel ownership verification framework CITED which is a first-of-its-kind method to achieve ownership verification on both embedding and label levels. Moreover, CITED is a novel signature-based method that neither harms downstream performance nor introduces auxiliary models that reduce efficiency, while still outperforming all watermarking and fingerprinting approaches. Extensive experiments demonstrate the effectiveness and robustness of our CITED framework. Code is available at: `https://anonymous.4open.science/r/CITED`.

## 1 Introduction

Graph Neural Network (GNN) (Kipf & Welling, 2016; Veličković et al., 2017; Hamilton et al., 2017a; Xu et al., 2018) has become a highly prevalent model, widely used in a variety of real-world applications, including recommendation systems (Ying et al., 2018; Berg et al., 2017) and molecular structure prediction (Gilmer et al., 2017; Feinberg et al., 2018). However, training such models, particularly at a large scale, is extremely difficult and costly (Duan et al., 2022; Bajaj et al., 2024), often requiring high-performance computing resources (AbdelBaky et al., 2012; Netto et al., 2018; Expósito et al., 2013; Gupta & Milojicic, 2011). As a result, these models represent valuable intellectual property, and model owners are reluctant to release them publicly. To balance usability and security, many developers adopt the Graph Machine Learning as a Service (GMLaaS) paradigm (Ribeiro et al., 2015; Weng et al., 2022; Zhang et al., 2020a), which exposes model functionalities through APIs while concealing the internal model structure. Several platforms provide such services in practice, including Amazon Neptune (Bebee et al., 2018), and Alibaba GraphScope (Xu et al., 2021). These GMLaaS systems offer convenient access to predictions for end users, while protecting the model's parameters and training data. However, these models are highly vulnerable to model extraction attacks (MEAs) (Liang et al., 2024; Zhang et al., 2021; Genç et al., 2023; Truong et al., 2021), in which an attacker can strategically query public APIs to obtain inference results and use these request-response pairs to train a model that mimics the functionalities of the original model. As a result, an attacker can obtain a surrogate model at a very low cost (Zhu et al., 2024), while still achieving functionality that is highly similar to the original model. Therefore, it is crucial to protect the valuable property of graph learning models by verifying the ownership of suspicious models that may have been obtained through MEA.

In practice, model owners may expose either predicted labels or intermediate embeddings to users, depending on the service interface. Therefore, ownership verification must be effective at both the

label and embedding levels to ensure comprehensive protection. To this end, recent efforts to defend against MEAs have focused on two main strategies (Regazzoni et al., 2021), watermarking (Adi et al., 2018; Liu et al., 2021) and fingerprinting (Cao et al., 2021; Wang & Chang, 2021). Watermarking techniques embed predefined input-output associations into the model, which can later be used to assert ownership by querying the model and matching responses (Boenisch, 2021; Szyller et al., 2021). However, it is difficult to extend watermarking to the embedding level, since embeddings are intermediate, continuous representations produced by the model, making it challenging to match them against specific target values for ownership verification. In contrast, fingerprinting leverages a model's unique internal characteristics, such as embedding distributions, to verify ownership (Chen et al., 2019; Cao et al., 2021). However, fingerprinting techniques are also ineffective at the label level, since prediction outputs reflect only the model's final decisions on downstream tasks, making it difficult to extract or compare intrinsic characteristics of the model from such discrete outputs. For example, injecting watermarks may alter the learned feature space and weaken the stability of fingerprints, while enforcing consistent embeddings for fingerprinting may suppress the variability required for watermark triggers. These conflicting objectives pose challenges for deploying both strategies simultaneously. Nevertheless, in GMLaaS environments, model owners may expose service interfaces at different output levels to accommodate diverse client requirements. As a result, MEA attackers can exploit these interfaces and launch extraction attacks at both the embedding and label levels. Therefore, protecting both label-level and embedding-level outputs is critical. The inability to achieve robust and unified protection across output levels exposes model owners to significant risk. As such, there is an urgent need for MEA defense frameworks that can operate reliably across diverse output levels.

Despite the significance of verifying model ownership across both embedding-level and label-level, developing a unified defense is non-trivial, and we mainly face three fundamental challenges. (1) **Ownership Indicator**: Crafting an ownership indicator that functions effectively across all output levels is exceptionally difficult, as the embedding level involves characterizing high-dimensional distributions, while the label level pertains to probabilistic predictions, making it hard to design a single verification mechanism that works well in both cases. (2) **Preservation of Ownership Indicators under MEAs**: Unlike backdoor attacks (Li et al., 2022), where adversaries have full access to the model, model extraction attacks (MEAs) require querying the target model to reconstruct a surrogate. This process is inherently more constrained, as adversaries typically rely on informative and natural inputs to maximize the utility of queries (Tramèr et al., 2016; Orekondy et al., 2019). Consequently, ownership indicators embedded through task-irrelevant triggers—such as watermarks (Zhao et al., 2021)—are unlikely to be activated or transferred to the surrogate model. Therefore, it is inherently difficult for attackers to successfully steal models that rely on ownership indicators under MEA settings. (3) **Efficiency of Verification**: In the ownership verification phase, it is common to train multiple auxiliary models to assist in distinguishing between genuine and unauthorized models, which introduces significant computational overhead. For example, fingerprinting-based techniques (Cao et al., 2021; You et al., 2024) often require training both positive and negative reference models to perform reliable verification. Such an approach leads to substantial resource consumption, making it inefficient and impractical for large-scale deployment in GMLaaS environments. Consequently, designing a unified and efficient defense mechanism that can verify model ownership under MEAs across all output levels remains a nascent and unresolved challenge.

To address these challenges, we propose *DeCision Boundary-Aware SIgnaTure for Model Extraction Attacks Defense* (CITED), a unified framework designed to provide effective defense against MEAs across all output levels. First, to tackle the challenge of *Ownership Indicator*, we introduce a novel signature-based ownership verification mechanism that is specifically tailored for MEA defense across all output levels. While the construction of the signature shares conceptual similarity with fingerprinting in terms of capturing model-specific characteristics, our verification paradigm is fundamentally different. Unlike watermarking, which relies on specially crafted input-output pairs, our signature serves as an intrinsic property of the model itself and does not require any task-irrelevant triggers. Second, to address the problem of *Preservation of Ownership Indicators under MEAs* in surrogate models, our method does not rely on post-training to embed input-output pairs. Instead, the signature naturally arises from the model's decision boundary. Since model extraction typically involves querying informative samples near the decision boundary (Brendel et al., 2017; He et al., 2018; Shen et al., 2023), these signature patterns are likely to be implicitly transferred into the surrogate model during the attack process, thereby enabling ownership verification without requiring explicit watermark triggers. Third, to improve the *Efficiency of Verification*, we propose an efficient verification protocol that avoids the need for training multiple auxiliary mod-

els, which is a common limitation in fingerprinting approaches. We extend the widely used ARUC metric (Cao et al., 2021; You et al., 2024) by incorporating the Wasserstein distance to better capture the distributional differences in signature embeddings between the target and suspect models. This Wasserstein-based ARUC metric enables more accurate and efficient ownership verification by querying the suspect model with signature inputs, especially under embedding-level access. Ultimately, our designed signature enables robust defense against MEAs across all output levels, setting it apart from prior methods. Our contributions are as follows:

- **Novel Problem Formulation:** We introduce a novel problem of GNN ownership verification by proposing a new type of ownership indicator, denoted as *signature*. Our proposed method enables verification at both the embedding and label levels of model output, effectively addressing limitations in prior works that focus on only one specific level.

- **Unified Defense Framework:** We propose CITED, a principled and unified framework for defending against graph-based model extraction attacks in the widely studied node classification task. It enables effective ownership verification by leveraging the proposed signature at both the embedding and label levels, with rigorous theoretical guarantees provided.

- **Comprehensive Experimental Evaluation:** We conduct extensive experiments demonstrating that our proposed method enables effective ownership verification at both the embedding and label levels. Compared to existing baselines, CITED consistently achieves significantly better performance, setting a new state-of-the-art in defending against MEAs.

## 2 PRELIMINARIES

### 2.1 NOTATIONS & THREAT MODEL

In this paper, we adopt the following notation conventions. We use lowercase bold symbols (e.g., $\boldsymbol{x}$) to denote vectors, and uppercase bold symbols (e.g., $\boldsymbol{X}$) to denote matrices. Calligraphic letters (e.g., $\mathcal{G}$) are used to represent sets or structured collections, such as graphs or datasets. Scalar values are denoted using lowercase italic symbols (e.g., $\tau$), and model functions are expressed as lowercase italic letters, such as $f$. We consider a graph $\mathcal{G} = (\mathcal{V}, \mathcal{E})$, where $\mathcal{V}$ denotes the set of nodes, $\mathcal{E} \subseteq \mathcal{V} \times \mathcal{V}$ denotes the set of edges. We define a graph-based model as a function $f : (\mathcal{V}, \mathcal{E}) \mapsto \mathcal{O}$, where the output set $\mathcal{O}$ varies depending on the output level of interest. The output set $\mathcal{O}$ varies depending on the level of supervision. At the embedding level, $\mathcal{O}^{\text{emb}} = \{\boldsymbol{h}_i \in \mathbb{R}^d : v_i \in \mathcal{V}\}$, where each $\boldsymbol{h}_i$ denotes the learned embedding of node $v_i$ in a latent feature space[1]. At the label level, $\mathcal{O}^{\text{label}} = \{y_i \in \{1, 2, \ldots, c\} : v_i \in \mathcal{V}\}$, where $y_i$ denotes the predicted class label associated with node $v_i$. For dataset collections denoted by $\mathcal{S} = ((\mathcal{V}, \mathcal{E}), \mathcal{O})$, we adopt the convention that superscripts specify the output level (e.g., emb or label), while subscripts indicate the functional role of the dataset (e.g., training, or querying). For example, $\mathcal{S}^{\text{emb}}_{\text{query}}$ refers to the dataset used by an attacker to query the model at the embedding level. We define several models to clarify our problem formulation. Specifically, $f_T$ represents the target model, which is trained and owned by the legitimate model proprietor. $f_I$ denotes an independent model, referring to a model trained separately by a third party using unrelated datasets and potentially different architectures. $f_S$ represents the surrogate model, which is illegitimately extracted by an attacker through MEAs. In addition, we denote by $f_D$ the defense model, which refers to the target model enhanced with ownership protection mechanisms. Finally, $f_Q$ denotes a suspicious model submitted for verification, which may potentially be a surrogate derived from the target model or an independently trained one.

**Definition 1** (Threat Model). *Given a target model $f_T$, a query graph $(\mathcal{V}_{query}, \mathcal{E}_{query})$, and access to the corresponding model outputs $\mathcal{O}^*_{query} = f_T(\mathcal{V}_{query}, \mathcal{E}_{query})$[2], the attacker aims to train a surrogate model $f_S$ that replicates the functional behavior of $f_T$ through model extraction attacks.*

In this paper, we focus on the widely studied transductive setting of model extraction attacks against GNNs (Genç et al., 2023; DeFazio & Ramesh, 2019), where the query graph satisfies $(\mathcal{V}_{\text{query}} \subseteq \mathcal{V}, \mathcal{E}_{\text{query}} \subseteq \mathcal{E})$. This setting naturally arises in many real-world applications such as citation networks (Kipf & Welling, 2016), social networks (Rossi et al., 2018), and recommendation systems (Ying et al., 2018), making it a practically significant scenario for MEA research.

---

[1] We consider the node embeddings as the output of the final graph convolutional layer in the GNN. These embeddings encode high-level node representations formed through message passing.

[2] The superscript $*$ indicates the output level, which can be either embedding (emb) or label (label) level.

## 2.2 PROBLEM FORMULATION

To defend against graph-based MEAs, existing ownership verification techniques primarily include watermarking and fingerprinting. Watermarking is mainly effective at the label level, but becomes impractical for continuous embedding. In contrast, fingerprinting usually captures distributional properties at the embedding level, yet these features do not generalize well to the label level due to limited output distinctiveness. To overcome these limitations, we introduce the concept of a *signature*, a unified ownership indicator applicable across both output levels.

**Definition 2** (Signature for GNNs). *A **signature** for GNNs is a graph-based dataset $\mathcal{S}_{sig} = ((\mathcal{V}_{sig}, \mathcal{E}_{sig}), \mathcal{O}_{sig})$, where $(\mathcal{V}_{sig}, \mathcal{E}_{sig}) \subseteq (\mathcal{V}, \mathcal{E})$ is a subgraph of the input graph, and $\mathcal{O}_{sig} \subseteq \mathcal{O}$ contains the corresponding outputs for $\mathcal{V}_{sig}$. Constructed near the decision boundary of the target model $f_T$, the signature captures predictive behaviors that are distinctive and sensitive to the model's parameters. Accessible only to the model owner, it serves as a compact, task-relevant identifier for ownership verification at both the embedding level ($\mathcal{O}^{emb}$) and label level ($\mathcal{O}^{label}$).*

Unlike watermarks and fingerprints, the *signature* is task-relevant, naturally aligned with the model's decision boundary, and effectively preserved in surrogate models through attacker queries. Its compatibility with both embedding and label outputs makes it a versatile and efficient tool for ownership verification. Based on this concept, we formulate a novel problem of ownership verification using signature sets across different output levels, as described below.

**Problem 1** (Ownership Verification with Signature for GNNs.). *Given a suspicious model $f_Q$, which may have been illicitly derived from a proprietary GMLaaS model $f_T$, and a signature dataset extracted from $f_T$, the goal is to determine whether $f_Q$ originates from $f_T$ by evaluating its behavior on the signature set. Specifically, the embedding-level signature $\mathcal{S}_{sig}^{emb}$ is used for verification at the embedding level, and the label-level signature $\mathcal{S}_{sig}^{label}$ is used for verification at the label level.*

## 3 METHODOLOGY

This section introduces the *DeCision Boundary-Aware SIgnaTure for Model Extraction Attacks Defense* (CITED), designed to provide effective protection across all output levels. We first introduce the generation of signature credentials, which extract boundary-based information as a unique identifier of model ownership. We then describe how model extraction attacks are carried out by adversaries and explain how our defense mechanism counteracts such attacks. Finally, we present a novel verification metric for validating ownership based on the extracted signature.

### 3.1 WORKFLOW OF THE PROPOSED SIGNATURE FRAMEWORK: CITED

**Pre-Attack Phase: Signature Generation.** The proposed framework allows model owners to adopt any Graph Machine Learning model $f_T$ appropriate for their downstream task, thereby highlighting the generalizability of our approach. To enable ownership verification, a signature generator module is employed to extract a unique signature associated with the target model. Formally, the signature dataset is defined as: $\mathcal{S}_{sig} = \Phi_{sig}((\mathcal{V}, \mathcal{E}), f_T(\mathcal{V}, \mathcal{E}))$, where $\Phi_{sig}$ denotes the *signature generator*. The resulting signature $\mathcal{S}_{sig} = ((\mathcal{V}_{sig}, \mathcal{E}_{sig}), \mathcal{O}_{sig}) \subseteq \mathcal{S}$ consists of a selected subgraph along with the corresponding outputs from the target model at the specified output level[3]. A detailed description of the signature construction process is provided in Section 3.2. The generated signature serves as a verifiable identifier for the target model and is later used to authenticate ownership and detect model misuse under model extraction attacks.

**Post-Attack Phase: Ownership Verification.** When a suspicious model $f_Q$ is identified, the model owner can perform ownership verification by leveraging the previously generated signature dataset $\mathcal{S}_{sig}$. Specifically, the owner queries the suspicious model $f_Q$ on the signature inputs $\mathcal{V}_{sig}$, obtaining outputs $\mathcal{O}_Q = f_Q(\mathcal{V}_{sig}, \mathcal{E}_{sig})$, and compares them to the reference outputs $\mathcal{O}_{sig} = f_T(\mathcal{V}_{sig}, \mathcal{E}_{sig})$ from the target model. To determine whether $f_Q$ originates from $f_T$, we compute a matching score between the two output sets at a specific output level. Formally, the matching is measured as: $M(\mathcal{O}_Q, \mathcal{O}_{sig})$, where $M$ denotes a level-specific matching metric quantifying the alignment between the outputs of the suspicious model and those of the target model, we will demonstrate it in

---

[3]For brevity, we omit the superscript $* \in \{\text{emb}, \text{label}\}$ in later expressions. Unless otherwise specified, symbols without superscripts are assumed to be general and applicable to either output level.

Section 5. If the matching score exceeds a predefined verification threshold $\sigma$, we conclude with high confidence that $f_Q$ is a surrogate derived from the proprietary model $f_T$.

## 3.2 Pre-Attack Phase: Signature Generation

To support ownership verification, we design a novel signature mechanism that differs fundamentally from watermarking and fingerprinting. Watermarking requires models to memorize trigger inputs with predefined outputs; however, such task-irrelevant patterns are hard to preserve under model extraction and impractical at the embedding level. Fingerprinting captures distributional features at the embedding level but fails to generalize to label-level verification and often incurs high computational cost due to auxiliary classifiers. In contrast, our signature directly leverages boundary-sensitive input-output pairs as credentials, enabling lightweight and accurate verification across output levels. Our model extraction attack and defense settings are based on widely accepted assumptions, which are discussed in detail in the Appendix C.1. We begin by identifying boundary nodes, and then construct the ownership-verification signature set $\mathcal{S}_{\text{sig}}$ based on these nodes using our proposed *signature score*.

**Boundary Nodes Identification.** We first introduce our decision boundary node generation method. Given a target model $f_T$, which performs node classification on a graph $(\mathcal{V}, \mathcal{E})$, we compute the logit matrix $\boldsymbol{Z} = f_T(\mathcal{V}, \mathcal{E}) \in \mathbb{R}^{|\mathcal{V}| \times c}$, where $\boldsymbol{Z}_{v,p}$ denotes the logit score assigned to node $v \in \mathcal{V}$ for class $p \in \{1, \ldots, c\}$. We then define a boundary voting score function $s_{\text{boundary}} : \mathcal{V} \to \mathbb{R}$, which assigns each node a score based on its logits. For each node $v \in \mathcal{V}$, let $p, q \in \{1, \ldots, c\}$ be the indices of the top-1, and top-2 largest entries in the vector $\boldsymbol{Z}_{v,:}$. The boundary score is defined as:

$$s_{\text{boundary}}(v) = \psi(\boldsymbol{Z}_{v,q} - \boldsymbol{Z}_{v,p}) - \lambda \cdot \mathcal{H}(\text{softmax}(\boldsymbol{Z}_{v,:})) \tag{1}$$

Here, $\psi$ denotes the ReLU function, which captures the gap between the two largest logit entries. And $\mathcal{H}(\cdot) = -\sum_{i=1}^{c} p_i \log p_i$ represents the entropy of the predicted class distribution, where $p_i = \text{softmax}(\boldsymbol{Z}_{v,:})_i$ denotes the predicted probability for class $i$. The purpose of the boundary score is to identify nodes that lie on or near the decision boundary. Accordingly, we compute the boundary score for all nodes $v \in \mathcal{V}$, and select the bottom $m\%$ with the lowest scores to form the boundary node set $\mathcal{S}_{\text{boundary}} \subseteq \mathcal{V}$. We extract the signature set $\mathcal{S}_{\text{sig}} \subseteq \mathcal{V}$ from the identified boundary nodes $\mathcal{S}_{\text{boundary}}$, as the decision boundary captures the most distinctive and model-specific behavior. Our objective is to select nodes near the boundary that reflect both classification ambiguity and unique decision patterns. To quantify these properties, we define a signature score composed of three complementary metrics: *boundary margin*, *boundary thickness*, and *heterogeneity*. These capture embedding-level distinctiveness, output-level uncertainty, and structural instability, respectively. The final signature score aggregates these metrics to guide the selection of informative, boundary-sensitive nodes (Yang et al., 2020).

**Boundary Margin.** At the embedding level, we evaluate the distinctiveness of model representations by measuring the proximity of each non-boundary node to its closest boundary node within the same class. We define the boundary margin score as:

$$s_{\text{margin}}(v_i, v_j) = \|\boldsymbol{h}_i - \boldsymbol{h}_j\|_2, \quad \text{for } v_i \in \mathcal{V} \setminus \mathcal{S}_{\text{boundary}}, \tag{2}$$

where $\boldsymbol{h}_i \in \mathbb{R}^d$ denotes the embedding of node $v_i$, and $y_i$ is the predicted label for $v_i$. A larger margin $s_{\text{margin}}$ indicates that boundary representations are more distinguishable from the rest of the graph, suggesting a clearer decision boundary. In contrast, smaller margin values imply greater ambiguity and highlight regions with greater discriminative sensitivity.

**Boundary Thickness.** At the label level, we assess classification ambiguity by comparing softmax outputs between each node $v_i \in \mathcal{V} \setminus \mathcal{S}_{\text{boundary}}$ and its closest boundary node $v_j \in \mathcal{S}_{\text{boundary}}$. Let $\boldsymbol{t}_i = \text{softmax}(\boldsymbol{z}_i) \in \mathbb{R}^c$ be the predicted probability vector of $v_i$, and $t_i = \boldsymbol{t}_i[y_i]$ its confidence in the predicted class $y_i = \arg\max \boldsymbol{t}_i$. The boundary thickness score is defined as:

$$s_{\text{thickness}}(v_i, v_j) = \|\boldsymbol{t}_i - \boldsymbol{t}_j\|_2 \cdot \sigma\left(\gamma - (t_i - t_j)\right), \tag{3}$$

where $\sigma(\cdot)$ is the sigmoid function, $\gamma$ is a confidence gap threshold. This score captures both prediction similarity and confidence gap, where larger values indicate weaker class separation and greater decision ambiguity near the boundary.

**Heterogeneity.** To capture structural uncertainty, we measure the label inconsistency in each node's local neighborhood. For each node $v_i \in \mathcal{V} \setminus \mathcal{S}_{\text{boundary}}$, the heterogeneity score is defined as:

$$s_{\text{hetero}}(v_i) = \frac{1}{|\mathcal{N}(v_i)|} \sum_{v_j \in \mathcal{N}(v_i)} \mathbf{1}\left[ y_i \neq y_j \right], \tag{4}$$

where $\mathcal{N}(v_i)$ is the set of 1-hop neighbors and $y_i$ is the predicted class label of $v_i$. This score quantifies local disagreement in predictions; higher values indicate structurally unstable regions likely to be near the decision boundary, where model-specific behaviors are most prominently expressed.

**Signature Score.** To identify the most informative nodes for ownership verification, we define an aggregated score $\hat{s}(v_i)$ for each node $v_i \in \mathcal{V} \setminus \mathcal{S}_{\text{boundary}}$, combining embedding distinctiveness, classification ambiguity, and structural uncertainty:

$$\hat{s}(v_i) = \min_{\substack{v_j \in \mathcal{S}_{\text{boundary}} \\ y_j = y_i}} \hat{s}_{\text{margin}}(v_i, v_j) + \alpha_1 \min_{\substack{v_j \in \mathcal{S}_{\text{boundary}} \\ y_j = y_i}} \hat{s}_{\text{thickness}}(v_i, v_j) - \alpha_2 \max_{v_i \in \mathcal{S}} \hat{s}_{\text{hetero}}(v_i), \tag{5}$$

where $\hat{s}_{\text{margin}}$, $\hat{s}_{\text{thickness}}$, and $\hat{s}_{\text{hetero}}$ are normalized scores, and $\alpha_1, \alpha_2 \geq 0$ are weighting coefficients. We then construct the signature set $\mathcal{S}_{\text{sig}}$ by selecting the bottom $\rho\%$ of nodes with the lowest aggregated scores: $\mathcal{S}_{\text{sig}} = \left\{ v_i \in \mathcal{V} \setminus \mathcal{S}_{\text{boundary}} \mid \hat{s}(v_i) \leq \tau \right\} \cup \mathcal{S}_{\text{boundary}}$, where $\tau$ is the $\rho$-quantile of the scores $\hat{s}(v_i)$. This ensures that selected nodes lie close to the decision boundary while capturing model-specific uncertainty across multiple levels.

### 3.3 POST-ATTACK PHASE: OWNERSHIP VERIFICATION

To verify model ownership, we design a unified evaluation framework that operates on a protected signature node set $\mathcal{S}_{\text{sig}} \subseteq \mathcal{V}$. The core intuition is that surrogate models—if extracted from a protected target—should exhibit output consistency with the target model on these signature nodes, while independently trained models should not. To quantify this separation, we adopt the *Area under the Robustness-Uniqueness Curve* (ARUC) (Cao et al., 2021), a widely used verification metric that captures the trade-off between recognizing surrogate models and rejecting unrelated ones. The full computation procedure is provided in Appendix C.2. We extend ARUC to both the embedding and label levels by designing level-specific matching scores that reflect output agreement between the suspicious model $f_Q$ and the target model $f_T$.

**Embedding level.** Each model produces high-dimensional embeddings on the signature nodes. We compute the 2-Wasserstein distance (Panaretos & Zemel, 2019) between the embeddings of the suspicious and target models:

$$M^{\text{emb}}(f_Q) = W_2\left( \mathcal{O}^{\text{emb}} f_Q, \mathcal{O}^{\text{emb}} \text{sig} \right). \tag{6}$$

Here, $W_2(\cdot, \cdot)$ denotes the 2-Wasserstein distance, which measures the optimal transport cost between two empirical distributions. We adopt $W_2$ as it effectively captures distributional differences between embedding spaces. In the context of MEAs, surrogate models often attempt to duplicate the target model by approximating its embedding distribution. Therefore, a smaller Wasserstein distance indicates closer alignment with the target model's behavior on the signature set, suggesting that the suspicious model may have inherited knowledge from the protected model.

**Label level.** Each model outputs discrete class predictions on the signature nodes. We compute the agreement as prediction accuracy with respect to the target model:

$$M^{\text{label}}(f_Q) = \frac{1}{|\mathcal{S}_{\text{sig}}|} \sum_{v \in \mathcal{S}_{\text{sig}}} \mathbf{1}\left[ \mathcal{O}^{\text{label}} f_Q(v) = \mathcal{O}^{\text{label}} \text{sig}(v) \right]. \tag{7}$$

Here, the matching score reflects the proportion of signature nodes for which the suspicious and target models produce the same predicted labels. In the context of model extraction attacks, adversaries often aim to duplicate the target model's decision boundaries. Since our signature nodes are specifically designed near such boundaries, surrogate models are more likely to match the target model's predictions on this subset. Therefore, a higher prediction accuracy indicates stronger behavioral alignment and suggests potential unauthorized model stealing from the target model.

## 4 THEORETICAL ANALYSIS

To establish a solid theoretical foundation for our ownership verification scheme, we model the attacker in a model extraction scenario as producing a surrogate GNN that differs from the target model by a small parameter perturbation. Prior work (Liao et al., 2020) shows that under such bounded perturbations, the output deviation of a message-passing GNN is also bounded. The formal statement of this result is provided in Lemma 3 in the Appendix. From Lemma 3 we have relative-perturbation ratio $\eta := \max_{i \in [L]} \frac{\|\boldsymbol{U}_i\|_2}{\|\boldsymbol{W}_i\|_2} \leq \frac{1}{L}$, which quantifies the layer-wise magnitude of parameter perturbation. Assuming zero-mean, independent, and almost surely bounded perturbations $\|\boldsymbol{U}_i\|_2 \leq \rho_i$, we define the proxy variance $\sigma^2 = (d\eta)^2 \left(\prod_{i=1}^{L-1}\|\boldsymbol{W}_i\|_2\right)^2 \sum_{i=1}^{L} \left(\frac{\rho_i}{\|\boldsymbol{W}_i\|_2}\right)^2$, which characterizes the overall perturbation scale. Building on Lemma 3 and Proposition 4, we now present the following theorem, which formalizes the theoretical guarantee of our ownership verification scheme under bounded parameter perturbations.

**Theorem 1** (Probabilistic Wasserstein Bound). *Let $\boldsymbol{e}_a = \boldsymbol{f_w}(\boldsymbol{X}, \boldsymbol{A})$ and $\boldsymbol{e}_b = \boldsymbol{f_{w+u}}(\boldsymbol{X}, \boldsymbol{A})$, where $\boldsymbol{u} = \mathrm{vec}(\{\boldsymbol{U}_1, \ldots, \boldsymbol{U}_L\})$. Suppose the random matrices $\{\boldsymbol{U}_i\}_{i=1}^{L}$ are mutually independent, satisfy $\mathbb{E}[\boldsymbol{U}_i] = \boldsymbol{0}$, and are almost surely bounded as $\|\boldsymbol{U}_i\|_2 \leq \rho_i$. For any $p \geq 1$, define $D_p = W_p(\delta_{\boldsymbol{e}_a}, \delta_{\boldsymbol{e}_b})$. If $D_p$ is sub-Gaussian with proxy variance $\sigma^2$, then for every $0 < \lambda < \Delta_{\mathcal{G}}$,*

$$\Pr(D_p < \lambda) \geq 1 - \exp\left(-\frac{(\Delta_{\mathcal{G}} - \lambda)^2}{2\sigma^2}\right). \tag{8}$$

*For all $\lambda \geq \Delta_{\mathcal{G}}$, the event $D_p \leq \Delta_G$ implies $\Pr(D_p < \lambda) = 1$.*

This theorem provides a theoretical guarantee for the effectiveness of the ownership verification method at the embedding level. Furthermore, based on Lemma 3 and Proposition 5, we also establish a theoretical foundation for ownership verification at the label level.

**Theorem 2** (Prediction-Agreement Probability). *Let $C$ be the number of classes. Let $\boldsymbol{z} = \boldsymbol{f_w}(\boldsymbol{X}, \boldsymbol{A}) \in \mathbb{R}^C$ denote the logit vector from the unperturbed model, and define $c^* = \arg\max_j \boldsymbol{z}_j$ as the predicted class with logit margin $\gamma = \boldsymbol{z}_{c^*} - \max_{j \neq c^*} \boldsymbol{z}_j > 0$. Let $\tilde{\boldsymbol{z}} = \boldsymbol{f_{w+u}}(\boldsymbol{X}, \boldsymbol{A})$ be the perturbed logits and $\tilde{c} = \arg\max_j \tilde{\boldsymbol{z}}_j$ the corresponding prediction. Then for any $p \geq 1$,*

$$\Pr(\tilde{c} = c^*) \geq 1 - (C - 1) \exp\left(-\frac{\gamma^2}{8\sigma^2}\right). \tag{9}$$

Consequently, CITED effectively leverages signature nodes as ownership indicators when surrogate models fall within the small-perturbation regime. These results show that surrogates generated by practical MEAs remain close in embedding space and aligned in label predictions with high probability. Our unified verification pipeline is designed to exploit these properties, offering strong theoretical support for CITED's effectiveness. Formal proofs are provided in Appendix B.

## 5 EXPERIMENTAL EVALUATIONS

In this section, we empirically evaluate the performance of the CITED framework. Specifically, we aim to address the following research questions: **RQ1**: How well does the CITED framework preserve the utility of the target model on downstream tasks? **RQ2**: How effective and efficient is CITED in verifying model ownership across different output levels? **RQ3**: What is the contribution of each component in the *signature* generation algorithm to the overall defense performance?

### 5.1 EXPERIMENTAL SETUP

**Downstream Tasks, Datasets and Backbone GNNs.** We evaluate the performance and generalizability of the proposed CITED framework on the fundamental graph learning task of node classification (Kipf & Welling, 2016). For this purpose, we adopt seven widely used graph datasets covering citation networks (Yang et al., 2016), e-commerce data (Shchur et al., 2018), and co-authorship graphs (Shchur et al., 2018). All dataset statistics and descriptions are provided in Appendix D.1. We conduct experiments using five representative backbone GNN models: GCN (Kipf & Welling,

Table 1: Performance evaluation of all defense methods on the original task, reported in terms of accuracy. All numerical values are in percentage. The best results are highlighted in **bold**.

| Model | Method | Cora | CiteSeer | PubMed | Photo | Computers | CS | Physics |
|---|---|---|---|---|---|---|---|---|
| GCN | GrOVe | $80.37 \pm 0.90$ | $68.97 \pm 0.55$ | $80.73 \pm 0.87$ | $92.80 \pm 0.20$ | $84.70 \pm 0.89$ | $\mathbf{91.27 \pm 0.57}$ | $89.23 \pm 1.36$ |
| | RandomWM | $80.10 \pm 0.44$ | $68.60 \pm 0.44$ | $80.63 \pm 0.50$ | $92.87 \pm 0.96$ | $83.50 \pm 2.72$ | $90.73 \pm 0.31$ | $90.40 \pm 2.51$ |
| | BackdoorWM | $77.97 \pm 0.60$ | $68.07 \pm 0.40$ | $79.27 \pm 0.49$ | $91.40 \pm 0.35$ | $81.60 \pm 1.11$ | $90.20 \pm 0.61$ | $90.57 \pm 0.80$ |
| | SurviveWM | $80.13 \pm 1.02$ | $68.10 \pm 1.47$ | $80.60 \pm 0.66$ | $91.60 \pm 0.53$ | $84.87 \pm 1.90$ | $90.83 \pm 0.85$ | $91.33 \pm 0.85$ |
| | CITED | $\mathbf{82.30 \pm 0.16}$ | $\mathbf{72.03 \pm 0.41}$ | $\mathbf{81.97 \pm 0.12}$ | $\mathbf{93.50 \pm 0.08}$ | $\mathbf{86.63 \pm 0.24}$ | $89.23 \pm 0.12$ | $\mathbf{94.47 \pm 0.12}$ |
| GAT | GrOVe | $82.00 \pm 0.62$ | $69.40 \pm 1.30$ | $79.63 \pm 0.85$ | $91.30 \pm 1.30$ | $84.93 \pm 1.71$ | $\mathbf{90.60 \pm 0.44}$ | $90.93 \pm 1.10$ |
| | RandomWM | $83.57 \pm 0.95$ | $70.23 \pm 0.90$ | $78.93 \pm 2.55$ | $\mathbf{92.13 \pm 0.49}$ | $\mathbf{87.00 \pm 0.36}$ | $90.13 \pm 0.85$ | $90.43 \pm 0.85$ |
| | BackdoorWM | $80.67 \pm 1.55$ | $68.97 \pm 1.56$ | $80.77 \pm 1.83$ | $90.97 \pm 0.46$ | $86.43 \pm 0.47$ | $88.17 \pm 0.55$ | $90.17 \pm 1.10$ |
| | SurviveWM | $82.80 \pm 1.32$ | $67.33 \pm 3.20$ | $77.10 \pm 2.86$ | $91.50 \pm 0.95$ | $86.77 \pm 1.14$ | $87.43 \pm 2.05$ | $84.10 \pm 5.40$ |
| | CITED | $\mathbf{84.10 \pm 1.02}$ | $\mathbf{70.77 \pm 0.37}$ | $\mathbf{79.80 \pm 0.22}$ | $90.87 \pm 2.29$ | $85.83 \pm 1.54$ | $90.50 \pm 0.22$ | $\mathbf{91.37 \pm 0.33}$ |
| SAGE | GrOVe | $82.40 \pm 0.52$ | $69.50 \pm 1.08$ | $78.80 \pm 1.11$ | $\mathbf{91.77 \pm 1.42}$ | $77.70 \pm 7.70$ | $91.87 \pm 0.50$ | $89.77 \pm 1.96$ |
| | RandomWM | $82.83 \pm 1.39$ | $69.77 \pm 0.29$ | $78.63 \pm 0.58$ | $91.73 \pm 0.23$ | $82.97 \pm 4.14$ | $\mathbf{91.97 \pm 0.76}$ | $90.33 \pm 0.64$ |
| | BackdoorWM | $80.87 \pm 0.15$ | $69.60 \pm 0.75$ | $77.57 \pm 0.90$ | $89.00 \pm 0.69$ | $72.60 \pm 2.31$ | $91.30 \pm 0.62$ | $89.73 \pm 0.40$ |
| | SurviveWM | $82.50 \pm 1.21$ | $70.47 \pm 0.85$ | $78.20 \pm 2.21$ | $91.50 \pm 0.17$ | $77.30 \pm 5.03$ | $91.23 \pm 0.55$ | $91.20 \pm 0.95$ |
| | CITED | $\mathbf{83.20 \pm 0.86}$ | $\mathbf{72.10 \pm 0.49}$ | $\mathbf{79.10 \pm 0.33}$ | $88.10 \pm 0.80$ | $\mathbf{84.00 \pm 2.11}$ | $91.07 \pm 0.12$ | $\mathbf{92.23 \pm 0.17}$ |
| GCNII | GrOVe | $79.67 \pm 0.85$ | $69.43 \pm 0.81$ | $80.27 \pm 1.48$ | $90.00 \pm 1.75$ | $85.53 \pm 2.18$ | $92.20 \pm 0.17$ | $91.60 \pm 1.41$ |
| | RandomWM | $79.60 \pm 0.95$ | $69.10 \pm 0.85$ | $80.70 \pm 0.78$ | $93.00 \pm 0.82$ | $\mathbf{85.90 \pm 1.61}$ | $91.70 \pm 0.17$ | $91.93 \pm 0.55$ |
| | BackdoorWM | $79.47 \pm 0.35$ | $68.93 \pm 0.86$ | $81.37 \pm 0.25$ | $92.70 \pm 0.92$ | $83.90 \pm 1.13$ | $90.40 \pm 0.96$ | $90.97 \pm 0.15$ |
| | SurviveWM | $80.83 \pm 1.00$ | $68.33 \pm 0.50$ | $81.07 \pm 0.55$ | $\mathbf{93.80 \pm 0.26}$ | $85.73 \pm 1.47$ | $91.93 \pm 0.74$ | $90.13 \pm 2.87$ |
| | CITED | $\mathbf{82.23 \pm 0.50}$ | $\mathbf{69.97 \pm 0.38}$ | $\mathbf{81.90 \pm 0.29}$ | $90.67 \pm 0.45$ | $83.80 \pm 1.20$ | $\mathbf{92.60 \pm 0.08}$ | $\mathbf{92.53 \pm 0.21}$ |
| FAGCN | GrOVe | $80.43 \pm 0.45$ | $71.33 \pm 0.15$ | $81.83 \pm 0.23$ | $93.23 \pm 0.42$ | $85.87 \pm 1.78$ | $92.67 \pm 0.38$ | $92.03 \pm 0.31$ |
| | RandomWM | $80.37 \pm 0.15$ | $\mathbf{71.60 \pm 0.53}$ | $81.20 \pm 0.10$ | $\mathbf{94.23 \pm 0.32}$ | $87.00 \pm 1.64$ | $92.67 \pm 0.12$ | $92.23 \pm 0.15$ |
| | BackdoorWM | $78.87 \pm 0.78$ | $70.03 \pm 1.02$ | $80.87 \pm 0.40$ | $93.10 \pm 0.10$ | $84.50 \pm 0.70$ | $91.83 \pm 0.49$ | $91.20 \pm 1.04$ |
| | SurviveWM | $80.60 \pm 0.85$ | $70.50 \pm 0.75$ | $\mathbf{82.27 \pm 0.81}$ | $94.17 \pm 0.21$ | $86.37 \pm 0.15$ | $91.97 \pm 0.59$ | $91.73 \pm 0.25$ |
| | CITED | $\mathbf{83.70 \pm 0.22}$ | $71.37 \pm 0.12$ | $81.20 \pm 0.08$ | $92.60 \pm 0.14$ | $\mathbf{83.70 \pm 0.33}$ | $\mathbf{92.73 \pm 0.12}$ | $\mathbf{92.60 \pm 0.16}$ |

2016), GAT (Veličković et al., 2017), and GraphSAGE (Hamilton et al., 2017b), as well as two state-of-the-art variants, GCNII (Chen et al., 2020) and FAGCN (Bo et al., 2021). Detailed architectural settings are summarized in Appendix F.4.

**Baselines, Threat Model.** To demonstrate the superiority of our CITED framework in defending against MEAs, we select state-of-the-art MEA defense methods in the graph domain as our baselines. These include watermark-based techniques such as RandomWM (Zhao et al., 2021), BackdoorWM (Xu et al., 2023), and SurviveWM (Wang et al., 2023), all of which operate at the label level to provide defense. Additionally, we include GrOVe (Waheed et al., 2023), a fingerprint-based method that performs defense at the embedding level. For the threat model, we adopt GNNStealing (Shen et al., 2022) to simulate practical model extraction attacks. The details of the baselines and the specific implementation of the threat model are provided in Appendix F.

**Evaluation Metrics.** To evaluate the performance of our CITED framework, we first assess its utility on the downstream task. Specifically, we adopt standard metrics including Accuracy, F1 Score, and AUROC to reflect the model's effectiveness on downstream task. In addition, we utilize the ARUC metric introduced in Section 3.3 to verify the effectiveness of ownership verification across different output levels. This metric has been widely adopted in related work (Cao et al., 2021; You et al., 2024). In addition, we report the AUC, a widely used metric that quantifies how well the verification scores can separate surrogate and extracted models (Li et al., 2019; Guan et al., 2022), with higher values indicating more reliable ownership verification. Details of the computation across different output levels used in our experiments are provided in Appendix C.3.

## 5.2 UTILITY OF CITED FRAMEWORK ON DOWNSTREAM TASKS

To address **RQ1**, we evaluate the performance of the defense model $f_D$ on the node classification task. We aim to investigate whether a defense model $f_D$ can maintain performance on the original task. As shown in Table 1, we conduct extensive experiments across various backbone models using different defense methods. We observe that watermark-based approaches generally suffer from performance degradation after the injection of watermark nodes. In contrast, our proposed method exhibits minimal degradation and, in some cases, even improves the model's performance. Notably, CITED consistently demonstrates highly competitive performance across most backbone models and datasets. Figure 1 further illustrates the effect of increasing the number of embedded ownership indicators. For watermarking methods, performance declines with more injected

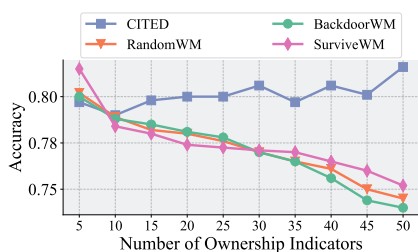

Figure 1: Performance of different defense models $f_D$ on the downstream task as the number of embedded ownership verification flags increases.

Table 2: Ownership verification performance of CITED compared to baselines at different output levels. Larger values indicate better effectiveness. The best results are highlighted in **bold**.

| Metric | Method | Cora | CiteSeer | PubMed | Photo | Computers | CS | Physics |
|---|---|---|---|---|---|---|---|---|
| ARUC$^{emb}$ | GrOVe | $55.87 \pm 0.47$ | $63.09 \pm 1.71$ | $37.09 \pm 0.69$ | $64.31 \pm 1.00$ | $65.09 \pm 1.20$ | $57.38 \pm 2.13$ | $48.04 \pm 1.12$ |
| | **CITED** | $\mathbf{62.20 \pm 0.63}$ | $\mathbf{66.51 \pm 1.14}$ | $\mathbf{62.76 \pm 1.95}$ | $\mathbf{65.62 \pm 1.52}$ | $\mathbf{65.80 \pm 1.23}$ | $\mathbf{70.76 \pm 1.76}$ | $\mathbf{61.71 \pm 1.24}$ |
| ARUC$^{label}$ | RandomWM | $16.51 \pm 4.65$ | $20.53 \pm 1.04$ | $15.04 \pm 2.51$ | $12.47 \pm 4.15$ | $17.47 \pm 8.64$ | $15.40 \pm 9.33$ | $24.98 \pm 2.11$ |
| | BackdoorWM | $32.84 \pm 23.2$ | $15.27 \pm 12.4$ | $16.13 \pm 1.04$ | $8.800 \pm 3.11$ | $19.07 \pm 5.77$ | $15.40 \pm 12.5$ | $16.13 \pm 13.5$ |
| | SurviveWM | $20.53 \pm 1.41$ | $18.44 \pm 1.82$ | $19.78 \pm 2.04$ | $33.73 \pm 5.19$ | $4.400 \pm 6.22$ | $37.22 \pm 3.08$ | $18.84 \pm 10.3$ |
| | **CITED** | $\mathbf{47.07 \pm 11.3}$ | $\mathbf{59.29 \pm 3.51}$ | $\mathbf{58.16 \pm 4.0}$ | $\mathbf{45.07 \pm 8.48}$ | $\mathbf{37.20 \pm 8.09}$ | $\mathbf{41.82 \pm 3.46}$ | $\mathbf{51.71 \pm 5.84}$ |

nodes. However, CITED remains robust, showing stable or slightly improved performance as more signature nodes are embedded. These results indicate that watermarking tends to harm model utility by injecting task-irrelevant nodes, while our signature nodes preserve task relevance and can even enhance learning. This highlights the practicality of CITED for real-world downstream applications. A more detailed analysis of utility performance is provided in Appendix E.1.

## 5.3 EFFECTIVENESS AND EFFICIENCY OF CITED FRAMEWORK

To address **RQ2**, we simulate a practical setting involving both MEAs and corresponding defenses. Specifically, we evaluate whether different defense methods can preserve ownership indicators in surrogate models while distinguishing them from independently trained models. Detailed settings and procedures are provided in Appendix D.2. As shown in Table 2, we evaluate the effectiveness of our method using the ARUC metric. Across all output levels, our proposed CITED framework achieves state-of-the-art results. Notably, at the label level, watermark-based

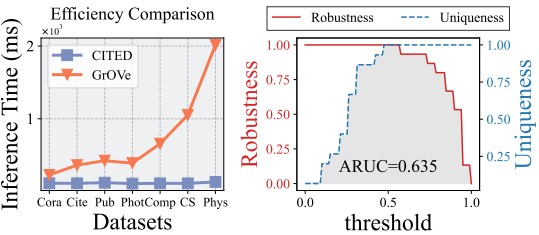

Figure 2: Left: Inference time comparison between CITED and GrOVe at the embedding level across varying datasets. Right: ARUC performance of our proposed signature for ownership verification.

methods perform poorly, highlighting the difficulty of preserving task-irrelevant ownership indicators through MEAs. Moreover, Figure 2 (right) presents the ARUC curve on the PubMed dataset, demonstrating that our approach achieves strong performance in both robustness and uniqueness. We also examine inference efficiency. As shown in Figure 2 (left), CITED is substantially faster than GrOVe, which depends on multiple auxiliary models and incurs significant overhead on large-scale datasets. Together, these results provide strong empirical evidence for the effectiveness and efficiency of CITED in ownership verification tasks. A more detailed analysis of verification effectiveness is provided in Appendix E.2.

## 5.4 ABLATION ANALYSIS OF SIGNATURE CREDENTIALS

To answer **RQ3**, we perform an ablation study to evaluate the contribution of each component in our signature scoring function. As shown in Table 3, we assess verification performance using signatures derived from individual components. Among them, the *thickness* score proves most effective, particularly at both the embedding and label levels, highlighting the importance of prediction ambiguity near decision boundaries. Integrating all components yields the highest verification accuracy, confirming that margin and complexity scores offer complementary value and validating the overall effectiveness of our signature design. A more detailed ablation study is provided in Appendix E.3.

Table 3: Effectiveness of individual components of the *signature score*, including boundary margin (M), boundary thickness (T), and heterogeneity (H), as well as their combined effect, evaluated at both the embedding level and the label level.

| Method | ARUC$^{emb}$ | ARUC$^{label}$ |
|---|---|---|
| CITED$_M$ | $69.67 \pm 2.11$ | $42.00 \pm 6.15$ |
| CITED$_T$ | $70.73 \pm 1.95$ | $53.60 \pm 16.0$ |
| CITED$_H$ | $66.00 \pm 1.14$ | $16.40 \pm 6.29$ |
| CITED$_{M+T+H}$ | $\mathbf{71.07 \pm 1.15}$ | $\mathbf{54.87 \pm 3.50}$ |

## 6 CONCLUSION

In this work, we propose CITED, a novel and unified framework for defending against model extraction attacks. Built upon decision boundary-aware signature nodes, CITED provides a theoretically grounded solution for ownership verification at both the embedding and label levels, effectively addressing the limitations of existing methods. Extensive experiments across diverse architectures and datasets demonstrate its superior verification effectiveness and efficiency, establishing CITED as a practical and effective approach for securing GNN models in real-world MLaaS settings.

ETHICS STATEMENT

Our work focuses on developing decision boundary-aware signatures to protect the intellectual property of graph neural networks (GNNs) against model extraction attacks. The datasets used in our experiments are all standard public benchmarks for graph and node classification tasks (e.g., Cora, Citeseer, PubMed, Amazon-Photo, Amazon-Computers, Coauthor-CS, Coauthor-Physics). These datasets contain only publicly available scientific or product-related information and do not involve sensitive personal data. Therefore, our study does not raise ethical concerns related to privacy or data misuse. We believe our contributions support responsible AI deployment by offering lightweight and effective tools for safeguarding GNN ownership in Machine Learning as a Service (MLaaS) settings.

REPRODUCIBILITY STATEMENT

We have taken deliberate steps to ensure the reproducibility of our results. All GNN backbone architectures are standard implementations from the `torch_geometric` library. For baseline watermarking and fingerprinting defenses, as well as model extraction attacks, we either rely on official code or carefully re-implement them according to the original papers, with detailed descriptions provided in Appendix F. We report all hyperparameters, dataset splits, training procedures, and evaluation protocols in the main text and appendix. Upon publication, we will release the complete source code, scripts, and configuration files to facilitate full reproducibility of our experiments.

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

## A  RELATED WORKS

**Graph-based Model Extraction Attacks & Defenses.** MEA is a prevalent threat faced by GM-LaaS. Due to the unique data structures inherent in graph machine learning (Xia et al., 2021; Stamile et al., 2021), existing MEA methods cannot be directly applied to graph models. Consequently, research on graph-based MEA remains in the early stages. The first paper about graph-based MEA (Wu et al., 2022) introduced a taxonomy comprising seven distinct attack scenarios, categorized by the extent of knowledge accessible to the attacker. Specifically, the availability or absence of knowledge regarding node attributes, graph structural information, and partial training data significantly impacts the performance of the surrogate model. Another work, GNNStealing (Shen et al., 2022), proposed a practical inductive graph stealing framework, categorizing attacks into two main types: Type I, in which attackers have knowledge of the graph structure of query data, and Type II, where such structural information is unavailable. Additionally, this work restricted the model outputs to align more closely with realistic scenarios, considering outputs such as node embeddings, soft labels, or node embeddings after dimensionality reduction via t-SNE (Van der Maaten & Hinton, 2008). Simultaneously, numerous methods have been proposed to defend against MEA. Among these, one early work (Zhao et al., 2021) introduced the idea of embedding watermarks into the target model using random graphs and specific outputs. Ownership can then be claimed if these watermarks are identified in a suspicious model. Furthermore, fingerprinting techniques (Waheed et al., 2023) determine model ownership by comparing the distributional proximity of suspicious embeddings with those of independent and target embeddings. In contrast to prior work, watermarking techniques require specific input-output pairs, limiting their applicability to the label level. Similarly, fingerprinting techniques rely on embedding distributions for verification, restricting their use to the embedding level. The CITED framework, however, introduces a unified approach that enables ownership verification across all output levels.

**Watermarking & Fingerprinting for GNNs.** Watermarking (Boenisch, 2021; Quiring et al., 2018; Singh & Singh, 2023; Zhang et al., 2020b) is a widely adopted technique to verify intellectual property (IP) ownership. In defending against MEA, task-specific watermarking approaches have been proposed for various graph-learning tasks. For the node classification task, the aforementioned work (Zhao et al., 2021) embeds watermarks by incorporating random graphs and assigning specific labels as outputs. For link prediction tasks, GENIE (Bachina et al., 2024) was proposed, embedding watermarks by modifying node features to follow specific distributions, altering subgraph structures, and producing predefined labels. Similarly, for graph classification tasks, another work (Xu et al.,

2023) embeds watermarks by adjusting subgraph structures and generating specific labels. More general watermarking techniques for GNNs have also addressed additional concerns. For instance, one recent work (Zhang et al., 2024) focuses explicitly on watermark imperceptibility and uniqueness. This approach demonstrates strong performance against evasion attacks, where surrogate models attempt to detect trigger graphs to evade generating desired outputs, thus invalidating ownership verification, as well as fraudulent attacks, in which adversaries forge watermarks to falsely claim model ownership. Additionally, fingerprinting-based defense methods have also been explored; for instance, GrOVe (Waheed et al., 2023) leverages a theorem stating that independently trained models naturally differ in their embedding distributions. By comparing whether the suspicious model's embedding distribution more closely aligns with the target model or an independently trained model, GrOVe determines if the suspicious model has been stolen from the target model. Compared with existing methods, our proposed CITED framework effectively preserves the signature in surrogate models even after MEA.

# B    PROOF OF THEORETICAL ANALYSIS

## B.1    LEMMA & PROPOSITIONS

**Lemma 3** (MPGNN Perturbation Bound (Liao et al., 2020)). *Let $L > 1$ be the number of layers and denote the parameter vector of an $L$-layer MPGNN by $\boldsymbol{w} = \mathrm{vec}\big(\{\boldsymbol{W}_1, \boldsymbol{W}_2, \ldots, \boldsymbol{W}_L\}\big) \in \mathbb{R}^m$, where each $\boldsymbol{W}_\ell \in \mathbb{R}^{d_\ell \times d_{\ell-1}}$ is the weight matrix of layer $\ell$. Consider an input feature matrix $\boldsymbol{X} \in \mathbb{R}^{|\mathcal{V}| \times d_0}$ that lies in $X_{R,\boldsymbol{h}_0} := \{ \boldsymbol{X} \mid \|\boldsymbol{X} - \boldsymbol{h}_0\|_2 \leq R \}$, with $\boldsymbol{h}_0 \in \mathbb{R}^{|\mathcal{V}| \times d_0}$ a fixed reference and $R > 0$ a radius. Let $\boldsymbol{A}$ be the adjacency matrix of a graph $\mathcal{G} = (\mathcal{V}, \mathcal{E})$ and let $d$ be its maximum node degree. For a perturbation $\boldsymbol{u} = \mathrm{vec}\big(\{\boldsymbol{U}_1, \boldsymbol{U}_2, \ldots, \boldsymbol{U}_L\}\big)$ satisfying $\eta := \max_{\ell \in [L]} \frac{\|\boldsymbol{U}_\ell\|_2}{\|\boldsymbol{W}_\ell\|_2} \leq \frac{1}{L}$, the output change of the MPGNN obeys*

$$\big\|\boldsymbol{f}_{\boldsymbol{w}+\boldsymbol{u}}(\boldsymbol{X}, \boldsymbol{A}) - \boldsymbol{f}_{\boldsymbol{w}}(\boldsymbol{X}, \boldsymbol{A})\big\|_2 \leq \mathrm{e}\, R\, L\, \eta\, \|\boldsymbol{W}_1\|_2\, \|\boldsymbol{W}_L\|_2\, C_\phi \frac{(dC)^{L-1} - 1}{dC - 1} = \Delta_{\mathcal{G}}$$

*where $C = C_\phi\, C_\rho\, C_g\, \|\boldsymbol{W}_2\|_2$ and $C_\phi, C_\rho, C_g$ are the Lipschitz constants of the activation function, the message aggregation operator, and the graph normalization operator, respectively.*

Based on Lemma 3, we conclude that for any MPGNN whose parameter perturbation $\boldsymbol{u}$ satisfies $\eta \leq 1/L$, the discrepancy between the perturbed model $\boldsymbol{f}_{\boldsymbol{w}+\boldsymbol{u}}$ and the original model $\boldsymbol{f}_{\boldsymbol{w}}$ is upper bounded by $\Delta_{\mathcal{G}}$. This bound depends on the radius $R$ of the input set, the perturbation ratio $\eta$, the spectral norms of the first and last weight matrices, the Lipschitz constants of the constituent operators, and the graph degree $d$. When $dC \neq 1$ the bound takes the closed form shown above, if $dC = 1$, then $\Delta_{\mathcal{G}} = \mathrm{e}\, R\, L\, \eta\, \|\boldsymbol{W}_1\|_2\, \|\boldsymbol{W}_L\|_2\, C_\phi\, (L - 1)$. A detailed proof is provided in the (Liao et al., 2020).

**Proposition 4** (Embedding Wasserstein Bound). *Let $\boldsymbol{e} = \boldsymbol{f}_{\boldsymbol{w}}(\boldsymbol{X}, \boldsymbol{A}) \in \mathbb{R}^d$ and $\tilde{\boldsymbol{e}} = \boldsymbol{f}_{\boldsymbol{w}+\boldsymbol{u}}(\boldsymbol{X}, \boldsymbol{A}) \in \mathbb{R}^d$ be the node-level embeddings produced by the original and the perturbed MPGNN, respectively. Interpret each embedding as a Dirac measure, $\delta_{\boldsymbol{e}}$ and $\delta_{\tilde{\boldsymbol{e}}}$, on $\mathbb{R}^d$. Then, for any $p \geq 1$,*

$$W_p\big(\delta_{\boldsymbol{e}}, \delta_{\tilde{\boldsymbol{e}}}\big) \leq \Delta_{\mathcal{G}},$$

*where $\Delta_{\mathcal{G}}$ is the perturbation bound established in Lemma 3.*

*Proof.* For Dirac measures, the $p$-Wasserstein distance collapses to the Euclidean distance between their support points, that is $W_p(\delta_{\boldsymbol{e}}, \delta_{\tilde{\boldsymbol{e}}}) = \|\boldsymbol{e} - \tilde{\boldsymbol{e}}\|_2$ for every $p \geq 1$. Lemma 3 guarantees $\|\boldsymbol{e} - \tilde{\boldsymbol{e}}\|_2 \leq \Delta_{\mathcal{G}}$, which directly implies the desired inequality. □

*Proof for Proposition 5.* The softmax map $\sigma : \mathbb{R}^C \to \Delta^{C-1}$ is 1-Lipschitz under the $\ell_2$ norm, i.e. $\|\sigma(u) - \sigma(v)\|_2 \leq \|u - v\|_2$ for all $u, v$. The maximum operator $\mathrm{conf}$ is 1-Lipschitz on the probability simplex with respect to the $\ell_2$ norm as well. Therefore

$$\big|\mathrm{conf}(p) - \mathrm{conf}(\tilde{p})\big| \leq \|p - \tilde{p}\|_2 \leq \|z - \tilde{z}\|_2 \leq \mathcal{B},$$

where the last inequality follows from Lemma 3. □

**Proposition 5** (Label Confidence Bound). *Let $z = f_w(X, A) \in \mathbb{R}^C$ and $\tilde{z} = f_{w+u}(X, A) \in \mathbb{R}^C$ be the logit vectors of a graph produced by the original and the perturbed MPGNN, respectively, where $u$ satisfies the hypothesis of Lemma 3. Set the soft labels to $p = \text{softmax}(z)$ and $\tilde{p} = \text{softmax}(\tilde{z})$. Define the prediction confidence as $\text{conf}(p) = \max_j p_j$. Then, for every graph input covered by Lemma 3,*

$$\left|\text{conf}(p) - \text{conf}(\tilde{p})\right| \leq \Delta_{\mathcal{G}}.$$

*Proof.* The soft-max map $\sigma : \mathbb{R}^C \to \Delta^{C-1}$ is 1-Lipschitz under the Euclidean norm, so $\|p - \tilde{p}\|_2 \leq \|z - \tilde{z}\|_2$. The maximum operator $\text{conf}$ is also 1-Lipschitz on the simplex, hence

$$\left|\text{conf}(p) - \text{conf}(\tilde{p})\right| \leq \|p - \tilde{p}\|_2 \leq \|z - \tilde{z}\|_2.$$

Lemma 3 bounds the right-most term by $\Delta_{\mathcal{G}}$, completing the proof. $\square$

## B.2 PROOF FOR THEOREM 1

*Proof of Theorem 1.* Recall that $D_p = W_p(\delta_{e_a}, \delta_{e_b}) = \|e_a - e_b\|_2$ for any $p \geq 1$. By the mean-value (Fréchet) expansion of $f_{w+u}$ around $w$,

$$e_b - e_a = \sum_{k=1}^{L} \left(\frac{\partial f}{\partial W_k}\right) U_k + \mathbf{R},$$

where $\mathbf{R} = O(\|u\|_2^2)$ collects second-order and higher terms.

**Bounding the remainder.** Lemma 3 imposes $\eta = \max_k \|U_k\|_2 / \|W_k\|_2 \leq 1/L$, so $\|u\|_2 = O(\eta)$. Consequently $\|\mathbf{R}\|_2 = O(\eta^2)$. Because $\eta \leq 1/L$ and $L \geq 2$, $\eta^2$ is at least one order smaller than the linear term and can be absorbed into the constant $\Delta_{\mathcal{G}}$; thus we focus on the linear part

$$\mathbf{S} = \sum_{k=1}^{L} \left(\frac{\partial f}{\partial W_k}\right) U_k.$$

**Sub-Gaussian tail for the linear term.** Each summand $\left(\frac{\partial f}{\partial W_k}\right) U_k$ is a mean-zero random vector that is independent across $k$ and has norm bounded by $d\,\eta\,\left(\prod_{i<k} \|W_i\|_2\right) \rho_k$, matching the components used in $\sigma^2$. Vector Bernstein (or equivalently the Hanson–Wright inequality applied coordinate-wise) therefore gives, for every $t \geq 0$,

$$\Pr\left(\|\mathbf{S}\|_2 \geq t\right) \leq \exp\left(-t^2 / (2\sigma^2)\right).$$

**Tail bound for $D_p$.** Since $\|\mathbf{R}\|_2 \leq \Delta_{\mathcal{G}} - \lambda$ for all $0 < \lambda \leq \Delta_{\mathcal{G}}$ whenever $\eta \leq 1/L$, the event $\|e_a - e_b\|_2 < \lambda$ is implied by $\|\mathbf{S}\|_2 < \Delta_{\mathcal{G}} - \lambda$. Hence, for $0 < \lambda \leq \Delta_{\mathcal{G}}$,

$$\Pr(D_p < \lambda) = \Pr\left(\|e_a - e_b\|_2 < \lambda\right) \geq \Pr\left(\|\mathbf{S}\|_2 < \Delta_{\mathcal{G}} - \lambda\right) \geq 1 - \exp\left(-\frac{(\Delta_{\mathcal{G}} - \lambda)^2}{2\sigma^2}\right).$$

**Deterministic regime.** Proposition 4 ensures $D_p \leq \Delta_{\mathcal{G}}$ almost surely, so $\Pr(D_p < \lambda) = 1$ for every $\lambda \geq \Delta_{\mathcal{G}}$.

Combining the two regimes completes the proof. $\square$

## B.3 PROOF FOR THEOREM 2

*Proof of Theorem 2.* Let $\tilde{z} - z = \delta$. By the Fréchet expansion of $f_{w+u}$ around $w$ we have

$$\delta = \sum_{k=1}^{L} \left(\frac{\partial f}{\partial W_k}\right) U_k + \mathbf{R}, \qquad \text{with } \|\mathbf{R}\|_2 = O(\eta^2).$$

Because $\eta \leq 1/L$ (Lemma 3), the remainder $\mathbf{R}$ is of smaller order than $\gamma$ and can be absorbed into the constant implicit in the tail bound below, so we focus on the linear term

$$\mathbf{S} = \sum_{k=1}^{L} \left( \frac{\partial \boldsymbol{f}}{\partial \boldsymbol{W}_k} \right) \boldsymbol{U}_k.$$

Each coordinate $S_j$ is a mean-zero sub-Gaussian random variable with proxy variance at most $\sigma^2$ defined in the Notation paragraph.

**Misclassification.** The predicted class changes from $c^*$ to some $\tilde{c} \neq c^*$ only if

$$S_{c^*} \leq -\tfrac{\gamma}{2} \quad \text{or} \quad S_j \geq \tfrac{\gamma}{2} \text{ for some } j \neq c^*.$$

**Tail probabilities.** For a sub-Gaussian variable $X$ with proxy variance $\sigma^2$, $\Pr(X \geq t) \leq \exp(-t^2/(2\sigma^2))$ and $\Pr(X \leq -t)$ obeys the same bound. Setting $t = \gamma/2$ yields

$$\Pr\left(S_{c^*} \leq -\tfrac{\gamma}{2}\right) \leq \exp\left(-\tfrac{\gamma^2}{8\sigma^2}\right), \qquad \Pr\left(S_j \geq \tfrac{\gamma}{2}\right) \leq \exp\left(-\tfrac{\gamma^2}{8\sigma^2}\right).$$

**Union bound.** Applying the union bound over the $(C-1)$ classes $j \neq c^*$ and the single negative-tail event for $c^*$,

$$\Pr(\tilde{c} \neq c^*) \leq (C-1) \exp\left(-\tfrac{\gamma^2}{8\sigma^2}\right) + \exp\left(-\tfrac{\gamma^2}{8\sigma^2}\right) = C \exp\left(-\tfrac{\gamma^2}{8\sigma^2}\right).$$

A slightly sharper bound, keeping only the $(C-1)$ positive-tail events, is $\Pr(\tilde{c} \neq c^*) \leq (C-1) \exp(-\gamma^2/(8\sigma^2))$; subtracting from one gives the stated inequality. $\qquad \square$

## C  SUPPLEMENTARY TECHNICAL DETAILS

### C.1  ASSUMPTIONS FOR MODEL EXTRACTION ATTACKS & DEFENSE

Our signature generation method is based on a widely accepted assumption that the decision boundary of an independently trained model is inherently unique (Cao et al., 2021; Waheed et al., 2023). These works provide both theoretical justification and empirical observations, demonstrating that this assumption holds broadly. Consequently, the design objective of our signature is to extract nodes that most effectively represent the boundary information as our signature.

**Assumption 6** (Decision Boundary Uniqueness). *Independently trained models, even when trained on the same dataset and architecture, possess unique output distributions and decision boundaries. Therefore, it is feasible to extract a signature that captures these boundaries and can serve as a unique identifier for model ownership verification.*

Additionally, we follow the assumption that attackers typically attempt to query information located at the decision boundary, which is widely adopted in other works (He et al., 2018; Brendel et al., 2017; Li et al., 2020; Shen et al., 2023). In Section 5, we further validate that if an attacker aims to more effectively replicate the functionality of the target model, they must query information near the decision boundary to achieve improved performance.

**Assumption 7** (Boundary Query Preference). *Adversarial model extraction attacks tend to query inputs near the decision boundary to maximize information gain, while benign users predominantly operate in high-confidence regions. Consequently, restricting access to boundary information can significantly degrade the effectiveness of surrogate models trained by attackers.*

To thoroughly analyze the effectiveness of our model, we require a more detailed definition of the attacker's capabilities. Specifically, an attacker cannot successfully replicate the behavior of a target model without accessing a sufficient amount of query-response data (Tramèr et al., 2016; Papernot et al., 2017).

**Assumption 8** (Minimum Effectiveness Query). *To train an effective surrogate model, an attacker needs sufficient query-response pairs from the target model. The surrogate's performance is limited by a minimum query budget. Below this threshold, the surrogate cannot reliably approximate the target model.*

## C.2 ARUC COMPUTATION FOR OWNERSHIP VERIFICATION

To quantify ownership verification performance, we employ the *Area under the Robustness-Uniqueness Curve* (ARUC) (Cao et al., 2021), which jointly considers the distinguishability of surrogate and independent models with respect to the signature node set $\mathcal{S}_{\text{sig}} \subseteq \mathcal{V}$. The evaluation proceeds in three steps: (1) computing a level-specific matching score $M(f_Q)$ between a suspicious model $f_Q$ and the target model $f_T$, (2) constructing robustness and uniqueness curves over a range of decision thresholds $\tau \in [0, 1]$, and (3) aggregating the two curves into a single ARUC value. Formally, let $F^+$ and $F^-$ denote the sets of surrogate (positive) and independent (negative) models, respectively. The robustness curve $R(\tau)$ and uniqueness curve $U(\tau)$ are defined as:

$$R(\tau) = \frac{1}{|F^+|} \sum_{f_Q \in F^+} \mathbf{1}[M(f_Q) \bowtie \tau], \quad U(\tau) = \frac{1}{|F^-|} \sum_{f_Q \in F^-} \mathbf{1}[M(f_Q) \bowtie' \tau],$$

Here, $\bowtie$ is a thresholding operator determined by the definition of the matching score at each output level, and $\bowtie'$ denotes its logical complement. Specifically, at the *embedding level*, the matching score is defined as the Wasserstein distance between the signature embeddings of the suspicious and target models. Since surrogate models are expected to lie closer in embedding space, the operator is set to $<$. In contrast, at the *label level*, the matching score measures the prediction accuracy on ownership indicators, where higher values reflect stronger alignment; hence, $\bowtie$ is set to $>$. Here, $\bowtie$ is a thresholding operator determined by the definition of the matching score at each output level, and $\bowtie'$ denotes its logical complement. Specifically, at the *embedding level*, the matching score is defined as the Wasserstein distance between the signature embeddings of the suspicious and target models. Since surrogate models are expected to lie closer in embedding space, the operator is set to $<$. In contrast, at the *label level*, the matching score measures the prediction accuracy on ownership indicators, where higher values reflect stronger alignment; hence, $\bowtie$ is set to $>$. Due to the nature of model extraction attacks, where surrogate models often exhibit moderate but not perfect alignment with the target, we normalize all matching scores across models before computing ARUC to ensure fair and comparable evaluation. To summarize the trade-off between robustness and uniqueness across thresholds, we compute the ARUC as:

$$\text{ARUC} = \frac{1}{r} \sum_{\tau'=1}^{r} \min\left\{ R\left(\frac{\tau'}{r}\right), U\left(\frac{\tau'}{r}\right) \right\},$$

where $r$ is the number of discrete threshold points sampled uniformly from the interval $[0, 1]$. A higher ARUC score indicates that the signature set enables strong alignment with surrogate models while effectively rejecting independent ones, thus ensuring reliable ownership verification. To summarize the trade-off between robustness and uniqueness across thresholds, we compute the ARUC as:

$$\text{ARUC} = \frac{1}{r} \sum_{\tau'=1}^{r} \min\left\{ R\left(\frac{\tau'}{r}\right), U\left(\frac{\tau'}{r}\right) \right\},$$

where $r$ is the number of discrete threshold points sampled uniformly from the interval $[0, 1]$. A higher ARUC score indicates that the signature set enables strong alignment with surrogate models while effectively rejecting independent ones, thus ensuring reliable ownership verification.

## C.3 AUC COMPUTATION FOR OWNERSHIP VERIFICATION

In our experiments, we adopt the *Area Under the Receiver Operating Characteristic Curve* (AUC) to evaluate the effectiveness of ownership verification. AUC quantifies the probability that a randomly selected surrogate model (i.e., a model suspected of being an unauthorized copy) yields a higher verification score than a randomly selected independent model. This provides an intuitive measure of how well the verification scores can distinguish surrogate models from independently trained ones. Formally, let $\{\mathbf{z}_i^+\}_{i=1}^n$ denote the outputs of suspicious surrogate models (positive samples), and $\{\mathbf{z}_j^-\}_{j=1}^m$ denote the outputs of independently trained models (negative samples). For each output $\mathbf{z}$, we define a score function $s(\mathbf{z}) = \text{score}(\mathbf{z}, \mathbf{z}_{\text{target}})$, which measures the agreement between $\mathbf{z}$ and the protected target model's output. The AUC is computed as:

$$\text{AUC} = \frac{1}{nm} \sum_{i=1}^n \sum_{j=1}^m \mathbb{1}\left[ s(\mathbf{z}_i^+) > s(\mathbf{z}_j^-) \right] \tag{10}$$

This corresponds to the Mann–Whitney U statistic, a standard formulation of AUC that captures the degree of separation between the score distributions of surrogate and independent models without requiring a fixed threshold. To account for tie cases where $s(\mathbf{z}_i^+) = s(\mathbf{z}_j^-)$, we apply averaging as follows:

$$\text{AUC} = \frac{1}{nm} \sum_{i=1}^{n} \sum_{j=1}^{m} \left[ \mathbb{1}(s(\mathbf{z}_i^+) > s(\mathbf{z}_j^-)) + \frac{1}{2} \cdot \mathbb{1}(s(\mathbf{z}_i^+) = s(\mathbf{z}_j^-)) \right] \tag{11}$$

The definition of the score function $s(\cdot)$ depends on the output level. At the *embedding level*, $\mathbf{z}$ represents the node embeddings of signature nodes, and the score $s(\mathbf{z})$ is calculated as the negative Wasserstein distance to the target model's embedding distribution. A higher score thus indicates stronger alignment with the target model. At the *label level*, $\mathbf{z}$ denotes the predicted class labels over the signature nodes, and the score $s(\mathbf{z})$ is defined as the classification accuracy with respect to the target model's predictions. In this case, a higher score reflects greater consistency in decision-making. By computing AUC independently at both the embedding and label levels, we obtain a unified yet adaptable metric to quantify how effectively surrogate models can be distinguished from non-infringing models across different output representations.

### C.4 TIME COMPLEXITY OF SIGNATURE GENERATION

The overall time complexity of signature generation is:

$$O(n \cdot B \cdot d + n \cdot k),$$

where $n$ is the number of nodes, $B$ is the number of boundary nodes, $d$ is the embedding dimension, and $k$ is the average node degree. The first term comes from computing the margin and thickness scores relative to boundary nodes, while the second term reflects the cost of evaluating local label heterogeneity. Given that $B \ll n$ and $k$ is small for sparse graphs, the procedure scales efficiently with graph size.

## D SUPPLEMENTARY EXPERIMENTAL SETTINGS

### D.1 EXPERIMENTAL DATA STATISTICS

Table 4 summarizes the statistics of the real-world graph datasets used in our experiments. These datasets are widely adopted in graph neural network (GNN) research, particularly for evaluating performance on node classification tasks. They cover a diverse range of domains, including citation networks (Cora, CiteSeer, PubMed), co-purchase graphs (Amazon-Photo, Amazon-Computers), and co-authorship networks (Coauthor-CS, Coauthor-Physics). All datasets represent realistic graph structures and have been extensively benchmarked in the literature.

Table 4: Statistics of the adopted real-world graph datasets.

|  | #Nodes | #Edges | #Attributes | #Classes |
|---|---|---|---|---|
| Cora | 2,708 | 5,429 | 1,433 | 7 |
| CiteSeer | 3,327 | 4,723 | 3,703 | 6 |
| PubMed | 19,717 | 88,648 | 500 | 3 |
| Amazon-Photo | 13,752 | 491,722 | 767 | 10 |
| Amazon-Computers | 7,650 | 238,162 | 745 | 8 |
| Coauthor-CS | 18,333 | 163,788 | 6,805 | 15 |
| Coauthor-Physics | 34,493 | 495,924 | 8,415 | 5 |

We carefully partition each dataset following the widely accepted protocol proposed in (Shchur et al., 2018), regenerating the splits to align with the model extraction attack (MEA) setting. Specifically, we sample 100 nodes per class for training. For validation and testing, we adopt the standard split sizes of 500 and 1000 nodes, respectively, as defined for Cora, CiteSeer, and PubMed. For

the Amazon and Coauthor datasets, which do not come with predefined splits, we apply the same protocol using equivalent validation and test set sizes.

### D.2 EXPERIMENTAL SETUP FOR OWNERSHIP VERIFICATION

We simulate a practical scenario involving both model extraction attacks (MEAs) and corresponding defense mechanisms. Specifically, we first apply various defense methods to protect the target model $f_D$. To emulate a realistic adversary, we construct 15 surrogate models $f_S$ trained using subgraph-query outputs from $f_D$, following typical attacker behavior under the assumed threat model.

Each surrogate model is built from one of five diverse backbone architectures and initialized with different random seeds. In parallel, we construct 15 independent models $f_I$, using the same five architectures and initialization strategy, but trained without querying $f_D$, instead following standard supervised learning procedures. This design enables a direct comparison between surrogate and independent models, allowing us to assess the extent to which different defenses preserve ownership signals in extracted models. All models are trained independently in accordance with our problem formulation and evaluation protocol.

Furthermore, for the attacker, we follow widely adopted settings to define the training protocol used during the model extraction attack (MEA). Specifically, at the embedding level, once the attacker obtains the embeddings, the surrogate model $f_S$ is trained by minimizing the MSE loss between its output and the query responses $O_{\text{query}}$. At the label level, under the commonly used setting where the attacker has access to the logits, we adopt the Knowledge Distillation approach (Hinton et al., 2015) to train the surrogate model $f_S$.

## E SUPPLEMENTARY EXPERIMENT RESULTS

### E.1 SUPPLEMENTARY EXPERIMENTS FOR UTILITY OF CITED FRAMEWORK

**Utility on Downstream Task.** In the main paper (Table 1), we evaluate task utility using classification accuracy as the primary metric. To further support these findings, we conduct a comprehensive set of experiments across five backbone models (GCN, GAT, GraphSAGE, GCNII, and FAGCN), five defense methods (GrOVe, RandomWM, BackdoorWM, SurviveWM, and CITED), and seven benchmark datasets (Cora, CiteSeer, PubMed, Amazon-Photo, Amazon-Computers, Coauthor-CS, and Coauthor-Physics). To provide a more complete view of model utility, we additionally report results using macro-F1 score and *Area Under the Receiver Operating Characteristic Curve* (AU-ROC) as alternative evaluation metrics. As shown in Table 8, CITED consistently achieves the best or highly competitive performance in terms of F1 score across most settings, closely matching the trends observed with accuracy. We further present AUROC results in Table 9, where CITED again demonstrates strong and stable performance across the majority of datasets and architectures. These supplementary evaluations confirm that CITED preserves task utility effectively under a wide range of conditions, reinforcing its reliability and practicality as an ownership verification method.

**Utility under Increasing Ownership Verification Indicators.** To complement the results originally presented in Section 5.2, which focused on the PubMed dataset, we extend the same experimental setting to six additional datasets: Cora, CiteSeer, Amazon-Photo, Amazon-Computers, Coauthor-CS, and Coauthor-Physics. As illustrated in Figure 3, we observe consistent trends across all these datasets. Specifically, watermarking-based defenses suffer from noticeable performance degradation as the number of injected nodes increases, which shows that adding task-irrelevant information can harm the model's performance. In contrast, CITED consistently maintains stable, or even slightly improved, task performance as more signature nodes are embedded. These supplementary results further confirm that watermarking approaches often compromise model utility, while CITED, by embedding signature nodes, successfully preserves and occasionally enhances downstream performance. This reinforces the practical utility of CITED across a wide range of datasets.

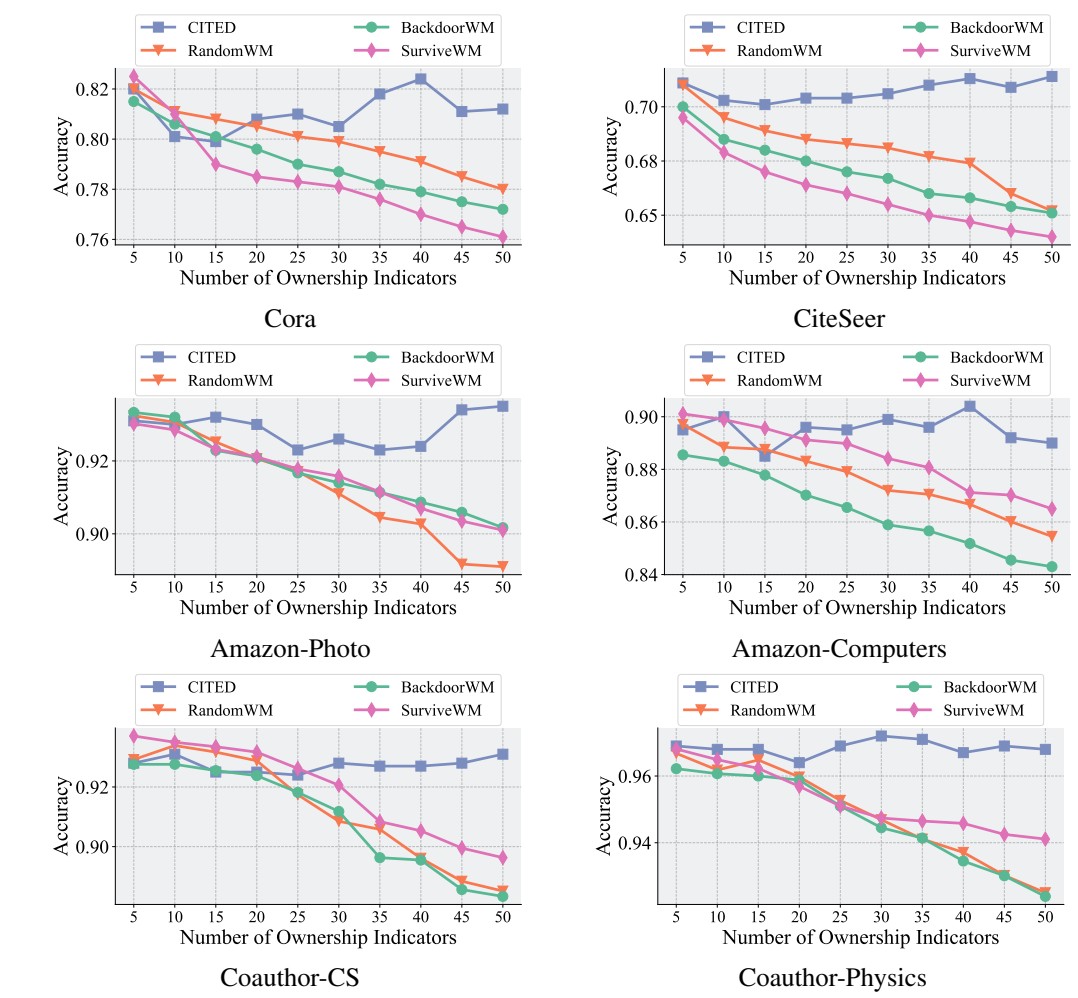

Figure 3: Performance of different defense models $f_D$ on the downstream task as the number of embedded ownership verification indicators increases.

### E.2 SUPPLEMENTARY EXPERIMENTS FOR EFFECTIVENESS AND EFFICIENCY OF CITED FRAMEWORK

**Robustness and Uniqueness Plot.** To further support the findings presented in Section 5.3, which focused on PubMed dataset, we extend the ARUC evaluation to six additional benchmark datasets: Cora, CiteSeer, Amazon-Photo, Amazon-Computers, Coauthor-CS, and Coauthor-Physics. The resulting evaluation curves are shown in Figure 4. Across all datasets, CITED consistently demonstrates strong performance in terms of both robustness against model extraction and uniqueness. The ARUC curves clearly indicate that the ownership indicators embedded by CITED are reliably retained in surrogate models while remaining distinguishable from those of unrelated models. These supplementary results provide further evidence for the reliability and effectiveness of the CITED framework in supporting ownership verification across diverse graph datasets.

**Robustness and Uniqueness Performance.** To further validate the effectiveness of our approach under varying architectural conditions, we replicate the ARUC evaluation using GCNII, a novel and more expressive backbone model. As shown in Table 5, CITED consistently achieves the highest ARUC scores at both the embedding and label levels across all datasets, confirming its ability to embed robust and distinctive ownership indicators that persist even under state-of-the-art GNN architectures. In contrast, watermark-based defenses continue to exhibit poor performance, particularly at the label level, highlighting their limited capacity to embed ownership information

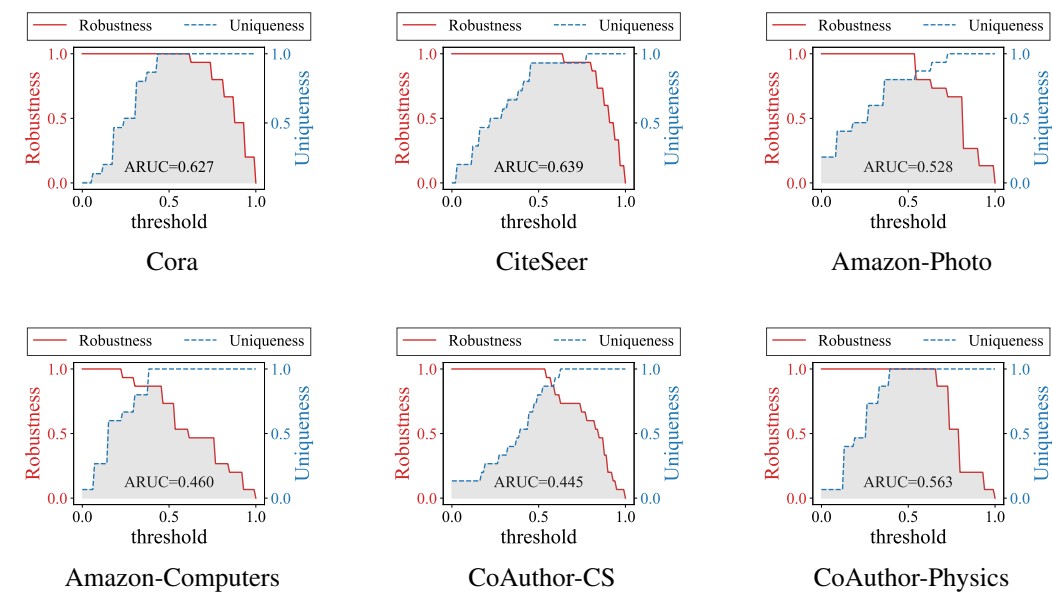

Figure 4: ARUC performance of our proposed signature for ownership verification.

Table 5: Ownership verification performance of CITED compared to baselines at different output levels. Larger values indicate better effectiveness.

| Metric | Method | Cora | CiteSeer | PubMed | Photo | Computers | CS | Physics |
|---|---|---|---|---|---|---|---|---|
| ARUC$^{emb}$ | GrOVe | 41.51 ± 1.64 | 48.96 ± 2.99 | 29.22 ± 1.41 | 52.18 ± 5.04 | 49.16 ± 3.55 | 52.38 ± 0.94 | 45.00 ± 0.33 |
| | **CITED** | **63.02 ± 0.91** | **58.58 ± 1.44** | **60.91 ± 0.31** | **63.11 ± 2.09** | **64.62 ± 2.61** | **59.62 ± 1.66** | **60.71 ± 1.78** |
| ARUC$^{label}$ | RandomWM | 13.80 ± 0.75 | 30.47 ± 8.01 | 29.13 ± 12.8 | 22.00 ± 15.6 | 19.07 ± 1.04 | 39.78 ± 9.46 | 28.60 ± 0.00 |
| | BackdoorWM | 25.91 ± 0.69 | 24.16 ± 1.85 | 20.82 ± 2.51 | 22.98 ± 3.40 | 12.20 ± 10.3 | 18.40 ± 10.5 | 21.02 ± 3.17 |
| | SurviveWM | 17.09 ± 7.67 | 23.02 ± 6.95 | 11.29 ± 0.53 | 18.38 ± 7.23 | 30.76 ± 12.6 | 35.00 ± 15.5 | 20.96 ± 5.55 |
| | **CITED** | **46.22 ± 5.15** | **59.76 ± 2.48** | **52.69 ± 3.54** | **31.16 ± 4.55** | **33.60 ± 8.70** | **35.60 ± 5.71** | **40.51 ± 5.34** |

that remains effective. These findings further reinforce the generalizability and reliability of the CITED framework across a range of modern GNN backbones.

**Distinguishability Performance.** To further assess model-level ownership verification ability under more advanced architectural settings, we evaluate the AUC by using GCNII as the backbone. The results are summarized in Table 6. We observe trends consistent with those reported in the main paper: at the embedding level, both CITED and GrOVe achieve ideal separability, indicating that the ownership signals embedded by these methods are well-preserved and easily distinguishable. More importantly, at the label level, CITED again significantly outperforms all baseline defenses, demonstrating a clear advantage in maintaining strong and transferable ownership indicators even under model extraction attacks. These supplementary findings further substantiate the robustness, effectiveness, and generalizability of the CITED framework in the context of ownership verification.

### E.3 Supplementary Experiments for Ablation Study of Signature Credentials

To complement the ablation results reported in Section 5.4, which focused on Cora dataset, we extend the evaluation of our signature design to six additional datasets: CiteSeer, PubMed, Amazon-Photo, Amazon-Computers, Coauthor-CS, and Coauthor-Physics, in addition to Cora. As summarized in Table 7, we observe consistent trends across all datasets. In most cases, particularly on CiteSeer, PubMed, and Coauthor-CS, the *thickness* component has the strongest individual impact, reaffirming its critical role in capturing decision-boundary ambiguity for ownership verification. Meanwhile, the *heterogeneity* component also demonstrates significant influence at the label level on Amazon-Photo and Amazon-Computers. On Coauthor-Physics, the *margin* component exhibits relatively stronger effects at the label level. These variations reflect the diverse and complex structural characteristics inherent to different datasets, which may affect the relative contributions of each component. Nevertheless, integrating all three components, margin, thickness, and heterogeneity,

Table 6: Effectiveness of each verification method under AUC. Larger AUC values indicate stronger separation between surrogate and independent models.

| Method | Cora | CiteSeer | PubMed | Photo | Computers | CS | Physics |
|---|---|---|---|---|---|---|---|
| GrOVe | $73.04 \pm 0.42$ | $79.70 \pm 3.09$ | $57.63 \pm 2.36$ | $92.44 \pm 6.33$ | $88.30 \pm 1.86$ | $90.52 \pm 3.04$ | $98.52 \pm 1.27$ |
| **CITED** | $100.0 \pm 0.00$ | $100.0 \pm 0.00$ | $100.0 \pm 0.00$ | $100.0 \pm 0.00$ | $100.0 \pm 0.00$ | $100.0 \pm 0.00$ | $100.0 \pm 0.00$ |
| RandomWM | $25.04 \pm 2.72$ | $50.52 \pm 4.82$ | $49.33 \pm 21.4$ | $16.30 \pm 11.5$ | $29.33 \pm 2.88$ | $70.52 \pm 12.2$ | $57.04 \pm 5.87$ |
| BackdoorWM | $50.22 \pm 4.40$ | $59.41 \pm 13.4$ | $39.11 \pm 5.03$ | $50.07 \pm 7.67$ | $27.56 \pm 27.4$ | $34.22 \pm 30.8$ | $38.96 \pm 7.12$ |
| SurviveWM | $41.63 \pm 18.7$ | $54.96 \pm 26.6$ | $24.59 \pm 1.47$ | $26.07 \pm 14.9$ | $48.59 \pm 23.2$ | $46.96 \pm 24.1$ | $34.67 \pm 4.12$ |
| **CITED** | $93.93 \pm 8.28$ | $99.85 \pm 0.21$ | $98.52 \pm 2.10$ | $75.70 \pm 10.14$ | $85.33 \pm 10.16$ | $85.48 \pm 7.82$ | $85.48 \pm 8.35$ |

typically yields the best verification performance, demonstrating that these components provide complementary information. These results confirm the robustness and effectiveness of our signature scoring design.

Table 7: Effectiveness of individual components of the *signature score*, including margin (M), thickness (T), and heterogeneity (H), as well as their combination. We report ARUC at both the embedding and label levels across six datasets.

| Method | CiteSeer | | PubMed | | Amazon-Photo | |
|---|---|---|---|---|---|---|
| | $ARUC^{emb}$ | $ARUC^{label}$ | $ARUC^{emb}$ | $ARUC^{label}$ | $ARUC^{emb}$ | $ARUC^{label}$ |
| $CITED_M$ | $71.60 \pm 1.77$ | $60.93 \pm 8.46$ | $65.20 \pm 3.96$ | $66.27 \pm 3.72$ | $72.73 \pm 1.11$ | $44.53 \pm 3.31$ |
| $CITED_T$ | $73.60 \pm 0.43$ | $70.33 \pm 2.03$ | $68.20 \pm 3.71$ | $66.80 \pm 2.32$ | $72.73 \pm 1.43$ | $45.73 \pm 4.98$ |
| $CITED_H$ | $73.13 \pm 1.06$ | $44.93 \pm 5.74$ | $65.20 \pm 2.73$ | $61.40 \pm 6.21$ | $69.00 \pm 1.42$ | $52.87 \pm 4.66$ |
| $CITED_{M+T+H}$ | $\mathbf{73.00 \pm 2.20}$ | $\mathbf{71.53 \pm 7.41}$ | $\mathbf{68.40 \pm 3.69}$ | $\mathbf{69.87 \pm 2.07}$ | $\mathbf{72.53 \pm 1.24}$ | $\mathbf{52.27 \pm 5.53}$ |

| Method | Amazon-Computers | | Coauthor-CS | | Coauthor-Physics | |
|---|---|---|---|---|---|---|
| | $ARUC^{emb}$ | $ARUC^{label}$ | $ARUC^{emb}$ | $ARUC^{label}$ | $ARUC^{emb}$ | $ARUC^{label}$ |
| $CITED_M$ | $72.07 \pm 1.64$ | $33.33 \pm 6.22$ | $77.40 \pm 0.59$ | $36.47 \pm 8.77$ | $73.87 \pm 3.27$ | $72.73 \pm 7.32$ |
| $CITED_T$ | $71.87 \pm 1.98$ | $31.47 \pm 3.40$ | $77.80 \pm 0.99$ | $37.67 \pm 8.22$ | $75.33 \pm 1.27$ | $60.87 \pm 9.51$ |
| $CITED_H$ | $68.20 \pm 1.82$ | $37.73 \pm 21.1$ | $75.60 \pm 1.84$ | $28.53 \pm 1.81$ | $68.53 \pm 0.77$ | $47.53 \pm 0.68$ |
| $CITED_{M+T+H}$ | $\mathbf{72.23 \pm 1.64}$ | $\mathbf{38.07 \pm 20.3}$ | $\mathbf{77.60 \pm 1.42}$ | $\mathbf{38.73 \pm 8.01}$ | $\mathbf{76.53 \pm 1.60}$ | $\mathbf{72.53 \pm 6.31}$ |

## E.4 ANALYSIS OF HYPERPARAMETER SENSITIVITY.

Our experiments demonstrate that the verification effectiveness is largely insensitive to hyperparameter choices as shown in Table 10. Across a wide range of settings, including variations in the coefficient distribution and noise level, the ARUC values remain stable with only marginal fluctuations. This robustness indicates that the verification mechanism does not rely on carefully tuned parameters, but instead captures intrinsic differences between surrogate and independent models. Consequently, practitioners can apply the method without exhaustive hyperparameter search, ensuring consistent reliability in practical deployments.

Table 8: Performance evaluation of all defense methods on the original task, reported in terms of F1-score. All numerical values are in percentage. Our results are highlighted in **bold**.

| Model | Method | Cora | CiteSeer | PubMed | Photo | Computers | CS | Physics |
|---|---|---|---|---|---|---|---|---|
| GCN | GrOVe | 80.18 ± 0.81 | 66.07 ± 0.35 | 80.10 ± 1.02 | 90.93 ± 0.12 | 82.19 ± 0.97 | 87.49 ± 1.03 | 88.05 ± 0.79 |
| | RandomWM | 79.83 ± 0.45 | 65.90 ± 0.57 | 79.94 ± 0.29 | 90.81 ± 0.88 | 79.13 ± 3.60 | 86.79 ± 0.93 | 88.96 ± 2.27 |
| | BackdoorWM | 77.85 ± 0.47 | 65.36 ± 0.50 | 78.43 ± 0.66 | 88.84 ± 0.26 | 79.16 ± 0.53 | 86.28 ± 0.44 | 88.50 ± 1.41 |
| | SurviveWM | 79.93 ± 0.99 | 65.31 ± 1.22 | 79.93 ± 0.55 | 89.58 ± 0.33 | 81.38 ± 0.92 | 86.99 ± 0.90 | 89.53 ± 0.44 |
| | **CITED** | **81.98 ± 0.13** | **68.92 ± 0.33** | **81.30 ± 0.14** | **91.42 ± 0.22** | **84.10 ± 0.39** | **85.82 ± 0.27** | **91.45 ± 0.25** |
| GAT | GrOVe | 81.42 ± 0.69 | 66.64 ± 1.10 | 78.80 ± 0.81 | 89.81 ± 1.77 | 82.85 ± 1.77 | 87.03 ± 0.11 | 89.31 ± 1.54 |
| | RandomWM | 82.87 ± 1.15 | 66.61 ± 0.58 | 77.84 ± 2.79 | 90.98 ± 0.46 | 84.34 ± 1.47 | 86.52 ± 0.75 | 88.62 ± 1.26 |
| | BackdoorWM | 80.64 ± 1.20 | 66.02 ± 1.15 | 79.66 ± 1.67 | 88.74 ± 0.44 | 83.78 ± 1.11 | 84.44 ± 0.45 | 87.79 ± 0.95 |
| | SurviveWM | 82.17 ± 1.37 | 65.07 ± 2.37 | 74.83 ± 2.48 | 90.05 ± 1.05 | 84.14 ± 1.75 | 83.42 ± 2.32 | 82.02 ± 3.70 |
| | **CITED** | **83.61 ± 0.89** | **67.59 ± 0.36** | **79.04 ± 0.22** | **89.30 ± 1.86** | **83.53 ± 2.25** | **87.82 ± 0.04** | **89.04 ± 0.27** |
| SAGE | GrOVe | 81.75 ± 0.41 | 66.30 ± 0.66 | 77.81 ± 0.96 | 90.08 ± 1.53 | 75.68 ± 6.44 | 88.84 ± 0.85 | 87.03 ± 2.11 |
| | RandomWM | 82.67 ± 1.29 | 66.69 ± 0.32 | 77.66 ± 0.46 | 90.23 ± 0.41 | 79.47 ± 4.58 | 88.88 ± 1.08 | 88.59 ± 1.04 |
| | BackdoorWM | 80.55 ± 0.60 | 66.77 ± 0.79 | 76.54 ± 0.76 | 87.08 ± 0.69 | 70.81 ± 0.85 | 88.73 ± 1.20 | 87.32 ± 0.79 |
| | SurviveWM | 82.27 ± 0.79 | 67.21 ± 0.95 | 77.29 ± 2.26 | 90.07 ± 0.39 | 73.72 ± 5.33 | 88.18 ± 1.17 | 89.29 ± 0.88 |
| | **CITED** | **82.61 ± 0.73** | **68.95 ± 0.47** | **78.13 ± 0.51** | **86.18 ± 0.72** | **80.87 ± 1.80** | **90.23 ± 0.16** | **89.77 ± 0.89** |
| GCNII | GrOVe | 79.38 ± 0.45 | 66.44 ± 0.68 | 79.64 ± 1.56 | 91.22 ± 1.90 | 84.00 ± 2.25 | 89.27 ± 0.62 | 90.14 ± 1.66 |
| | RandomWM | 79.54 ± 0.96 | 66.44 ± 0.82 | 80.08 ± 0.81 | 91.32 ± 0.82 | 83.21 ± 1.19 | 87.92 ± 0.62 | 90.53 ± 0.48 |
| | BackdoorWM | 78.94 ± 0.28 | 66.25 ± 0.59 | 80.45 ± 0.56 | 90.39 ± 1.26 | 79.90 ± 1.59 | 86.14 ± 1.14 | 88.73 ± 0.82 |
| | SurviveWM | 80.41 ± 0.93 | 65.70 ± 0.50 | 80.55 ± 0.42 | 92.06 ± 0.19 | 83.11 ± 1.66 | 88.57 ± 1.08 | 87.00 ± 2.98 |
| | **CITED** | **82.37 ± 0.39** | **67.26 ± 0.31** | **81.12 ± 0.29** | **88.69 ± 0.27** | **80.79 ± 1.54** | **89.37 ± 0.11** | **90.88 ± 0.17** |
| FAGCN | GrOVe | 80.24 ± 0.40 | 68.21 ± 0.19 | 81.17 ± 0.23 | 91.57 ± 0.25 | 84.06 ± 1.61 | 90.03 ± 0.46 | 90.95 ± 0.50 |
| | RandomWM | 80.24 ± 0.13 | 68.55 ± 0.74 | 80.52 ± 0.13 | 92.59 ± 0.32 | 85.00 ± 1.81 | 89.93 ± 0.14 | 91.43 ± 0.15 |
| | BackdoorWM | 78.93 ± 0.64 | 67.31 ± 1.22 | 80.15 ± 0.32 | 90.55 ± 0.21 | 81.65 ± 0.79 | 88.83 ± 0.75 | 90.33 ± 0.80 |
| | SurviveWM | 80.32 ± 0.80 | 67.71 ± 0.71 | 81.80 ± 0.83 | 92.45 ± 0.21 | 84.80 ± 0.61 | 88.72 ± 0.87 | 89.75 ± 0.54 |
| | **CITED** | **83.44 ± 0.23** | **68.42 ± 0.13** | **80.69 ± 0.12** | **90.72 ± 0.05** | **81.79 ± 0.38** | **89.45 ± 0.19** | **91.27 ± 0.31** |

Table 9: Performance evaluation of all defense methods on the original task, reported in terms of AUROC. All numerical values are in percentage. Our results are highlighted in **bold**.

| Model | Method | Cora | CiteSeer | PubMed | Photo | Computers | CS | Physics |
|---|---|---|---|---|---|---|---|---|
| GCN | GrOVe | 96.65 ± 0.15 | 87.06 ± 0.31 | 93.38 ± 0.16 | 99.53 ± 0.08 | 98.64 ± 0.10 | 99.38 ± 0.08 | 98.09 ± 0.17 |
| | RandomWM | 96.69 ± 0.02 | 87.06 ± 0.07 | 93.33 ± 0.23 | 99.55 ± 0.04 | 98.63 ± 0.26 | 99.37 ± 0.08 | 98.33 ± 0.33 |
| | BackdoorWM | 95.85 ± 0.32 | 86.30 ± 0.20 | 91.55 ± 0.53 | 97.90 ± 0.30 | 97.63 ± 0.44 | 98.79 ± 0.13 | 98.10 ± 0.32 |
| | SurviveWM | 96.80 ± 0.06 | 86.73 ± 0.34 | 93.28 ± 0.39 | 99.35 ± 0.17 | 98.66 ± 0.03 | 98.99 ± 0.22 | 98.59 ± 0.22 |
| | **CITED** | **97.40 ± 0.00** | **89.19 ± 0.14** | **94.01 ± 0.06** | **99.59 ± 0.00** | **99.09 ± 0.02** | **99.49 ± 0.01** | **99.82 ± 0.01** |
| GAT | GrOVe | 97.50 ± 0.11 | 87.45 ± 0.65 | 93.26 ± 0.35 | 99.30 ± 0.12 | 98.90 ± 0.08 | 99.32 ± 0.02 | 98.33 ± 0.18 |
| | RandomWM | 97.42 ± 0.14 | 88.42 ± 0.36 | 93.64 ± 0.25 | 99.44 ± 0.07 | 98.92 ± 0.04 | 99.28 ± 0.05 | 98.58 ± 0.25 |
| | BackdoorWM | 96.74 ± 0.17 | 87.83 ± 0.71 | 92.49 ± 0.55 | 98.44 ± 0.34 | 97.36 ± 0.34 | 98.66 ± 0.17 | 98.24 ± 0.12 |
| | SurviveWM | 97.40 ± 0.27 | 87.82 ± 0.38 | 93.29 ± 0.85 | 99.45 ± 0.04 | 98.99 ± 0.07 | 98.94 ± 0.21 | 96.77 ± 0.58 |
| | **CITED** | **97.79 ± 0.03** | **89.21 ± 0.05** | **93.80 ± 0.07** | **99.37 ± 0.06** | **98.97 ± 0.10** | **99.46 ± 0.02** | **98.92 ± 0.07** |
| SAGE | GrOVe | 97.31 ± 0.06 | 88.28 ± 0.48 | 92.70 ± 0.12 | 99.28 ± 0.17 | 97.89 ± 0.89 | 99.10 ± 0.19 | 97.75 ± 0.80 |
| | RandomWM | 97.34 ± 0.21 | 87.84 ± 0.44 | 92.60 ± 0.47 | 99.21 ± 0.01 | 97.91 ± 0.70 | 99.05 ± 0.23 | 98.10 ± 0.15 |
| | BackdoorWM | 96.93 ± 0.24 | 87.64 ± 0.13 | 91.02 ± 0.47 | 97.64 ± 0.58 | 95.52 ± 0.50 | 98.62 ± 0.43 | 97.33 ± 0.47 |
| | SurviveWM | 97.28 ± 0.22 | 88.79 ± 0.66 | 91.63 ± 0.13 | 99.13 ± 0.14 | 97.48 ± 0.67 | 98.74 ± 0.30 | 98.50 ± 0.18 |
| | **CITED** | **97.71 ± 0.05** | **90.28 ± 0.07** | **92.65 ± 0.07** | **99.02 ± 0.02** | **98.67 ± 0.13** | **99.41 ± 0.00** | **99.08 ± 0.02** |
| GCNII | GrOVe | 96.58 ± 0.11 | 87.73 ± 0.46 | 93.31 ± 0.13 | 99.54 ± 0.07 | 98.64 ± 0.15 | 99.53 ± 0.02 | 98.68 ± 0.15 |
| | RandomWM | 96.78 ± 0.10 | 87.69 ± 0.01 | 93.27 ± 0.24 | 99.56 ± 0.04 | 98.77 ± 0.10 | 99.47 ± 0.13 | 98.77 ± 0.08 |
| | BackdoorWM | 96.20 ± 0.22 | 87.02 ± 0.35 | 91.73 ± 0.31 | 98.39 ± 0.20 | 97.45 ± 0.19 | 98.70 ± 0.08 | 98.35 ± 0.16 |
| | SurviveWM | 96.81 ± 0.09 | 87.21 ± 0.50 | 93.03 ± 0.30 | 99.58 ± 0.03 | 98.68 ± 0.13 | 99.36 ± 0.05 | 98.80 ± 0.13 |
| | **CITED** | **97.45 ± 0.01** | **88.57 ± 0.09** | **93.98 ± 0.06** | **99.46 ± 0.03** | **98.95 ± 0.03** | **99.62 ± 0.01** | **99.21 ± 0.02** |
| FAGCN | GrOVe | 97.37 ± 0.06 | 89.83 ± 0.20 | 93.79 ± 0.12 | 99.46 ± 0.09 | 98.94 ± 0.02 | 99.57 ± 0.06 | 99.25 ± 0.02 |
| | RandomWM | 97.32 ± 0.05 | 89.45 ± 0.12 | 93.70 ± 0.14 | 99.57 ± 0.04 | 98.92 ± 0.04 | 99.58 ± 0.02 | 99.24 ± 0.02 |
| | BackdoorWM | 96.85 ± 0.05 | 88.76 ± 0.14 | 91.72 ± 0.09 | 97.85 ± 0.17 | 97.56 ± 0.14 | 98.95 ± 0.19 | 98.67 ± 0.34 |
| | SurviveWM | 96.86 ± 0.18 | 89.41 ± 0.19 | 93.83 ± 0.25 | 99.55 ± 0.03 | 98.97 ± 0.06 | 99.41 ± 0.13 | 98.83 ± 0.24 |
| | **CITED** | **97.88 ± 0.02** | **89.65 ± 0.01** | **94.02 ± 0.01** | **99.48 ± 0.02** | **99.00 ± 0.01** | **99.61 ± 0.00** | **99.27 ± 0.00** |

Table 10: ARUC results under different coefficient settings. Results are reported as mean ± std.

| coefficient | ARUC$^{emb}$ | ARUC$^{label}$ |
|---|---|---|
| (0.300, 0.40, 0.300) | 67.87 ± 0.01 | 45.33 ± 0.04 |
| (0.275, 0.45, 0.275) | 66.24 ± 0.02 | 46.76 ± 0.06 |
| (0.250, 0.50, 0.250) | 67.73 ± 0.01 | 50.40 ± 0.06 |
| (0.225, 0.55, 0.225) | 68.02 ± 0.04 | 47.55 ± 0.09 |
| (0.200, 0.60, 0.200) | 66.07 ± 0.02 | 55.27 ± 0.10 |
| (0.175, 0.65, 0.175) | 66.23 ± 0.03 | 50.58 ± 0.16 |
| (0.150, 0.70, 0.150) | 68.27 ± 0.02 | 51.47 ± 0.17 |
| (0.125, 0.75, 0.125) | 69.36 ± 0.02 | 50.61 ± 0.13 |
| (0.100, 0.80, 0.100) | 71.13 ± 0.02 | 55.73 ± 0.06 |
| (0.075, 0.85, 0.075) | 70.40 ± 0.01 | 50.14 ± 0.06 |
| (0.050, 0.90, 0.050) | 69.60 ± 0.02 | 52.27 ± 0.10 |
| (0.025, 0.95, 0.025) | 70.83 ± 0.01 | 51.27 ± 0.09 |

## E.5 Robustness Against Removal Attacks

To further validate this, we conducted dedicated experiments where we applied additional removal attacks after the surrogate model was constructed via model extraction. The experimental results will be presented subsequently. Specifically, we considered:

- Pruning (Zhang et al., 2018): Removing 30% of the weights in the surrogate model;
- Fine-tuning (Rouhani et al., 2018): Updating the surrogate model using unseen data;
- Distribution shift (Orekondy et al., 2019): Querying the target model with inputs perturbed by Gaussian noise in the feature space.

These settings—pruning, fine-tuning, and distribution shift—are widely acknowledged evaluation settings in the literature for testing the robustness of watermarking or verification schemes under model extraction scenarios. In all cases, we used our signature-based verification scheme to detect ownership. As shown in the Table 11, while the performance of CITED does degrade slightly, particularly on larger datasets such as Coauthor-CS and Coauthor-Physics, it consistently outperforms all baseline methods under these challenging conditions.

Table 11: Performance of robustness under removal attack. Results are reported as mean $\pm$ std.

| Method | Cora | CiteSeer | PubMed | Photo | Computers | CS | Physics |
|---|---|---|---|---|---|---|---|
| RandomWM | 31.56 ± 1.26 | 58.67 ± 13.2 | 30.67 ± 2.51 | 21.48 ± 4.82 | 16.30 ± 9.91 | 22.67 ± 9.43 | 62.07 ± 12.1 |
| BackdoorWM | 43.85 ± 31.0 | 23.85 ± 16.1 | 27.49 ± 5.44 | 4.890 ± 3.14 | 35.70 ± 11.1 | 18.52 ± 10.5 | 29.93 ± 24.1 |
| SurviveWM | 41.19 ± 2.30 | 39.56 ± 3.14 | 37.04 ± 0.84 | 48.89 ± 6.29 | 1.780 ± 2.51 | 46.37 ± 7.96 | 37.33 ± 17.6 |
| CITED | **89.04 ± 7.76** | **99.85 ± 0.21** | **98.67 ± 1.89** | **90.52 ± 3.02** | **88.15 ± 5.63** | **97.93 ± 0.91** | **98.24 ± 0.36** |
| CITED_prune | **88.00 ± 0.10** | **94.18 ± 0.22** | **89.24 ± 0.09** | **82.67 ± 0.15** | **83.68 ± 0.07** | **56.00 ± 0.12** | **67.87 ± 0.14** |
| CITED_fintune | **77.33 ± 0.11** | **97.33 ± 0.04** | **90.67 ± 0.07** | **81.33 ± 0.11** | **85.33 ± 0.04** | **66.53 ± 0.17** | **64.17 ± 0.10** |
| CITED_shift | **84.00 ± 0.09** | **91.76 ± 0.05** | **88.65 ± 0.11** | **82.67 ± 0.17** | **82.00 ± 0.06** | **54.27 ± 0.04** | **65.25 ± 0.09** |

These findings demonstrate that CITED *retains strong robustness* even under removal attacks, and highlight its suitability for ownership verification in practical, adversarial environments.

## E.6 Robustness under Noisy Labels and Class Imbalance

In this subsection, we present an extended evaluation of the robustness of the signature mechanism under two challenging but realistic conditions, namely noisy labels and severe class imbalance. All experiments follow the same pipeline as described in Section 5. We first train the target model, apply CITED to identify and finetune the signature nodes, and then perform ownership verification against both independent models and extraction based surrogate models. The experimental results for both settings are reported below.

**Noisy Labels.** To investigate the impact of corrupted annotations, we introduce label noise by randomly flipping a specified proportion of training labels while keeping all other components unchanged. The ownership verification results for the label level are summarized in Table 12. Across all datasets and noise ratios, CITED maintains a stable level of accuracy for both the embedding level and the label level. This demonstrates that the extracted signature nodes remain distinctive even when the training labels contain substantial noise. Surrogate models tend to preserve the decision boundary related characteristics of the target model, enabling CITED to reliably separate them from independently trained models.

Table 12: Ownership verification performance under noisy labels at the label level.

| Noise Ratio | Cora | Citeseer | PubMed | Photo | Computers | CS | Physics |
|---|---|---|---|---|---|---|---|
| 0.1 | 0.3193±0.0273 | 0.5428±0.0000 | 0.5180±0.0024 | 0.4743±0.0249 | 0.4381±0.0286 | 0.6109±0.0203 | 0.7108±0.0000 |
| 0.3 | 0.4100±0.1556 | 0.5580±0.0122 | 0.5330±0.0000 | 0.4920±0.0000 | 0.4513±0.0105 | 0.6577±0.0000 | 0.7043±0.0152 |
| 0.5 | 0.4633±0.0603 | 0.5640±0.0436 | 0.5764±0.0001 | 0.5313±0.0014 | 0.4400±0.0229 | 0.6482±0.0187 | 0.7521±0.0109 |
| 0.7 | 0.5107±0.0585 | 0.6141±0.0142 | 0.5451±0.0000 | 0.5140±0.0164 | 0.4263±0.0116 | 0.6602±0.0148 | 0.7846±0.0001 |
| 0.9 | 0.7160±0.0368 | 0.6615±0.0213 | 0.5523±0.0117 | 0.5062±0.0146 | 0.4128±0.0281 | 0.6280±0.0110 | 0.7474±0.0020 |

These results demonstrate that CITED is resistant to a wide range of noisy label intensities and that the decision boundary related signature pattern remains reliably detectable.

**Class Imbalance.** We further evaluate the robustness of CITED under class imbalance, a common scenario in real world datasets. Given an imbalance ratio, we flip the labels of a specified fraction of minority class nodes into the majority class. Tables 13 and 14 report the results for the label level and embedding level, respectively.

Table 13: Ownership verification performance under class imbalance at the label level.

| Imbalance Ratio | Cora | Citeseer | PubMed | Photo | Computers | CS | Physics |
|---|---|---|---|---|---|---|---|
| 0.1 | 0.6953±0.0019 | 0.1840±0.1265 | 0.6460±0.0057 | 0.4893±0.0443 | 0.4107±0.0442 | 0.5767±0.0560 | 0.7280±0.0205 |
| 0.3 | 0.7080±0.0057 | 0.1840±0.1301 | 0.6900±0.0000 | 0.7287±0.0038 | 0.7760±0.0033 | 0.5913±0.0019 | 0.6753±0.0160 |
| 0.5 | 0.7073±0.0038 | 0.0000±0.0000 | 0.7600±0.0000 | 0.9460±0.0000 | 0.9333±0.0009 | 0.8473±0.0019 | 0.6900±0.0000 |
| 0.7 | 0.7053±0.0009 | 0.9520±0.0000 | 0.9160±0.0000 | 0.9780±0.0000 | 0.9687±0.0025 | 0.9540±0.0000 | 0.8040±0.0000 |
| 0.9 | 0.6887±0.0019 | 0.9560±0.0000 | 0.9540±0.0000 | 0.9760±0.0000 | 0.9667±0.0034 | 0.9760±0.0000 | 0.9140±0.0000 |

Table 14: Ownership verification performance under class imbalance at the embedding level.

| Imbalance Ratio | Cora | Citeseer | PubMed | Photo | Computers | CS | Physics |
|---|---|---|---|---|---|---|---|
| 0.1 | 0.6953±0.0019 | 0.6980±0.0028 | 0.6700±0.0000 | 0.7033±0.0084 | 0.6640±0.0049 | 0.7320±0.0000 | 0.6913±0.0009 |
| 0.3 | 0.7080±0.0057 | 0.6960±0.0102 | 0.6840±0.0000 | 0.6940±0.0173 | 0.6707±0.0038 | 0.7320±0.0000 | 0.6980±0.0000 |
| 0.5 | 0.7073±0.0038 | 0.7053±0.0038 | 0.6860±0.0000 | 0.6920±0.0118 | 0.6713±0.0038 | 0.7360±0.0000 | 0.7040±0.0000 |
| 0.7 | 0.7053±0.0009 | 0.7140±0.0000 | 0.6760±0.0000 | 0.6953±0.0118 | 0.6647±0.0038 | 0.7360±0.0000 | 0.7140±0.0000 |
| 0.9 | 0.6887±0.0019 | 0.7140±0.0000 | 0.6660±0.0000 | 0.7053±0.0147 | 0.6687±0.0050 | 0.7200±0.0000 | 0.7100±0.0000 |

A notable observation is that the separation between surrogate models and independent models becomes even more pronounced as the imbalance ratio grows. This is consistent with observations made in prior work, which report that independent models trained under imbalance tend to exhibit highly divergent decision boundary shapes, while extraction based surrogate models remain closely aligned with the target model near the boundary region. This amplifies the discriminative power of CITED.

### E.7 COMPRESSION AND STRUCTURAL OPTIMIZATION OF THE SIGNATURE SET

In this subsection, we investigate whether the signature set produced by CITED can be compressed or structurally optimized without compromising ownership verification reliability. We consider two complementary directions. The first direction examines whether a reduced subset of signature nodes can still preserve strong verification performance. The second direction explores whether the commitment that the model owner publishes can be compressed for practical deployment, while maintaining full verification correctness.

**Grouped Signature Compression.** To study the effect of reducing the size of the signature set, we first generate the complete signature set using the full CITED pipeline, and then apply a KMeans based clustering procedure to group signature nodes by their embedding similarity. From each cluster, we select the most representative node, forming a grouped signature super set. The grouping ratio specifies the fraction of signature nodes retained after clustering. Tables 15 and 16 report the verification results at the label level and embedding level, respectively.

Table 15: Ownership verification performance under grouped signature compression at the label level.

| Group Ratio | Cora | Citeseer | PubMed | Photo | Computers | CS | Physics |
|---|---|---|---|---|---|---|---|
| 0.25 | 0.4231±0.0224 | 0.5100±0.0028 | 0.5793±0.0637 | 0.4266±0.0150 | 0.4619±0.0262 | 0.5486±0.0191 | 0.4380±0.0001 |
| 0.50 | 0.4369±0.0316 | 0.5360±0.0318 | 0.6500±0.0172 | 0.4406±0.0899 | 0.4626±0.0423 | 0.5060±0.0662 | 0.4646±0.0339 |
| 0.75 | 0.4660±0.0112 | 0.5986±0.0530 | 0.6540±0.0299 | 0.4706±0.0188 | 0.5286±0.0228 | 0.5513±0.0217 | 0.5306±0.0627 |
| 1.00 | 0.4817±0.0413 | 0.5607±0.0377 | 0.6540±0.0197 | 0.4793±0.1148 | 0.5893±0.0586 | 0.5586±0.0247 | 0.5560±0.0056 |

The results indicate that CITED remains highly reliable even when the signature set is significantly reduced. Performance degradation becomes noticeable only when the grouping ratio falls below approximately 0.3, which is expected because aggressively removing nodes reduces redundancy. Importantly, even with substantial compression, the performance consistently exceeds that of baseline verification methods. These observations confirm that the signature nodes identified by CITED contain sufficient uniqueness to allow structural reduction without compromising verification effectiveness.

Table 16: Ownership verification performance under grouped signature compression at the embedding level.

| Group Ratio | Cora | Citeseer | PubMed | Photo | Computers | CS | Physics |
|---|---|---|---|---|---|---|---|
| 0.25 | 0.6298±0.0149 | 0.6486±0.0009 | 0.6326±0.0150 | 0.6593±0.0009 | 0.6720±0.0000 | 0.6673±0.0024 | 0.6280±0.0000 |
| 0.50 | 0.6445±0.0065 | 0.6293±0.0037 | 0.6413±0.0041 | 0.6688±0.0028 | 0.6860±0.0016 | 0.6573±0.0108 | 0.6313±0.0018 |
| 0.75 | 0.6779±0.0031 | 0.7094±0.0043 | 0.6619±0.0043 | 0.6793±0.0018 | 0.7060±0.0016 | 0.6933±0.0024 | 0.7093±0.0037 |
| 1.00 | 0.6748±0.0000 | 0.7060±0.0000 | 0.6773±0.0000 | 0.6680±0.0016 | 0.7080±0.0016 | 0.6999±0.0001 | 0.6940±0.0000 |

**Hash Based Commitment Compression.** Beyond reducing the number of signature nodes, we further explore compressing the commitment that the model owner must publish for verification. The commitment consists of the indices of the selected signature nodes. For space efficient deployment, we concatenate the node indices into a byte sequence and compute a sixty four bit hash digest. During verification, the model owner reveals the signature node indices, the verifier recomputes the digest, and ownership is confirmed if the hash matches the published commitment and the suspicious model produces a high similarity score on the revealed nodes.

Table 17 reports the original commitment size and the compressed size for different signature set ratios. The hash based approach achieves very large compression savings while preserving exact verification validity.

Table 17: Compression of the signature commitment using a sixty four bit hash digest.

| Sig Set Ratio | Original Size (bytes) | Compressed Size (bytes) | Compression Saving (%) |
|---|---|---|---|
| 0.0247 | 292 | 8 | 97.26 |
| 0.0288 | 345 | 8 | 97.68 |
| 0.0354 | 427 | 8 | 98.12 |
| 0.0576 | 692 | 8 | 98.84 |
| 0.0819 | 987 | 8 | 99.18 |

These results demonstrate that commitment compression through hashing is extremely effective and introduces no loss in verification reliability. This enables lightweight publication and transmission of ownership evidence in real world applications.

### E.8 MINIMUM SIGNATURE SIZE FOR STABLE VERIFICATION

In this subsection, we study how the size of the signature set affects ownership verification performance. This analysis provides insight into the minimum number of signature nodes needed to ensure reliable verification and examines whether this threshold varies with graph scale or model capacity. To this end, we vary the signature set ratio, which determines the fraction of signature nodes used during verification, and evaluate the resulting performance across all benchmark datasets.

Table 18: Ownership verification performance when varying the signature set ratio.

| Sig Set Ratio | Cora | Citeseer | PubMed | Photo | Computers | CS | Physics |
|---|---|---|---|---|---|---|---|
| 0.0247 | 0.4140±0.0000 | 0.6480±0.0000 | 0.6960±0.0000 | 0.6213±0.0344 | 0.4823±0.0114 | 0.6220±0.0000 | 0.7440±0.0000 |
| 0.0288 | 0.4420±0.0000 | 0.6080±0.0000 | 0.5840±0.0000 | 0.2920±0.0000 | 0.4853±0.0075 | 0.6120±0.0000 | 0.7440±0.0000 |
| 0.0354 | 0.4600±0.0000 | 0.6640±0.0000 | 0.6740±0.0001 | 0.4313±0.0090 | 0.4460±0.0129 | 0.6520±0.0000 | 0.7560±0.0000 |
| 0.0576 | 0.4681±0.0002 | 0.6340±0.0000 | 0.6400±0.0000 | 0.3020±0.0254 | 0.5033±0.0179 | 0.6800±0.0000 | 0.8600±0.0001 |
| 0.0819 | 0.4940±0.0056 | 0.6713±0.0857 | 0.6953±0.0047 | 0.5026±0.0249 | 0.3980±0.0412 | 0.6613±0.0603 | 0.7686±0.0122 |

The results in Table 18 show that verification performance remains strong for a wide range of signature ratios. A mild degradation becomes noticeable only when the signature set ratio falls below approximately $0.3$, which is expected since extremely small signature sets reduce redundancy in the verification process. However, even in these low ratio settings, the performance of CITED consistently exceeds that of all baseline methods.

Based on these observations, we regard a ratio of around $0.3$ as a practical threshold that balances verification reliability with computational and storage efficiency. Importantly, this threshold remains stable across different graph scales and model capacities in our experiments, indicating that the boundary related signature nodes selected by CITED exhibit strong generality.

### E.9 GENERALIZATION ABILITY

In this subsection, we demonstrate that CITED can be generalized beyond node classification to support graph classification tasks and regression models. Only minimal modifications are required to adapt the signature extraction procedure to these additional settings, and CITED maintains strong verification performance across all evaluated tasks. In the following experiments, we adopt a more general and widely adopted knowledge distillation based threat model, which captures a broad class of extraction attacks and applies uniformly across different learning paradigms.

**Graph Classification.** For graph classification, the prediction target corresponds to an entire graph instead of individual nodes. To adapt the heterogeneity component of our method, we replace the one hop neighborhood comparison with a nearest neighbor based measure, which is directly applicable to independent identical distribution settings that commonly arise in graph level tasks. Using this modification, we apply CITED to two widely used benchmarks, namely ENZYMES and PROTEINS. The results are shown in Table 19.

Table 19: Ownership verification results for graph classification tasks.

| Model | Metric | ENZYMES | PROTEINS |
|-------|--------|---------|----------|
| CITED | ARUC | $0.4277\pm0.0535$ | $0.3747\pm0.1162$ |
| CITED | AUC | $1.0000\pm0.0000$ | $0.9778\pm0.0143$ |

The results show that CITED consistently provides strong separation between surrogate models and independently trained models. In particular, CITED achieves one hundred percent AUC on ENZYMES and near perfect AUC on PROTEINS, confirming that signature based verification remains highly effective at the graph level.

**Regression.** To examine whether our method extends to continuous valued prediction tasks, we further evaluate CITED on the widely used molecular regression dataset ZINC. Table 20 reports the corresponding results.

Table 20: Ownership verification results for the regression task on the ZINC dataset.

| Model | Metric | ZINC |
|-------|--------|------|
| CITED | ARUC | $0.5164\pm0.0191$ |
| CITED | AUC | $1.0000\pm0.0000$ |
| CITED | RMSE | $1.7349\pm0.0112$ |

These results demonstrate that CITED reliably differentiates surrogate models from independent models even in regression settings, achieving perfect AUC and strong ARUC performance. The RMSE of the target model remains stable after incorporating the signature finetuning stage, indicating that the defense does not negatively affect predictive utility.

Taken together, the experiments across classification and regression tasks show that CITED generalizes effectively across different learning paradigms.

### E.10 COMPLEXITY ANALYSIS

In this subsection, we provide a detailed analysis of the computational complexity of the proposed signature generation procedure, followed by empirical measurements of its runtime on multiple benchmark datasets. These results confirm that our method introduces only minimal overhead compared to the standard training cost of graph neural networks.

**Theoretical Complexity.** Let $n$ denote the number of nodes, $B$ the number of boundary nodes selected during the signature extraction process, $d$ the embedding dimension, and $k$ the average node degree. The total time complexity of generating the signature set is

$$\mathcal{O}(n \cdot B \cdot d + n \cdot k).$$

The term $n \cdot B \cdot d$ corresponds to computing the margin and thickness scores with respect to the selected boundary nodes. The term $n \cdot k$ arises from evaluating local label heterogeneity by inspecting the neighborhood of each node. Since typical benchmark graphs satisfy $B \ll n$ and have a small degree $k$ due to sparsity, the overall procedure scales efficiently with graph size.

**Empirical Runtime Comparison.** To complement the theoretical analysis, we benchmark the actual runtime of signature generation across seven widely used datasets and compare it with the full training time of a standard GCN model with 200 epochs. Table 21 summarizes the results. The signature extraction is consistently efficient and contributes less than five percent of end to end model training time.

Table 21: Runtime of signature generation compared to standard GCN training. All results are reported as mean $\pm$ standard deviation over multiple trials.

|  | Cora | CiteSeer | PubMed | Photo | Computers | CS | Physics |
|---|---|---|---|---|---|---|---|
| Sig Gen Time (s) | $0.05 \pm 0.06$ | $0.05 \pm 0.06$ | $0.05 \pm 0.05$ | $0.04 \pm 0.06$ | $0.04 \pm 0.05$ | $0.04 \pm 0.05$ | $0.05 \pm 0.06$ |
| Training Time (s) | $1.18 \pm 0.01$ | $1.04 \pm 0.09$ | $2.70 \pm 0.26$ | $4.09 \pm 1.10$ | $4.76 \pm 0.01$ | $6.54 \pm 0.02$ | $19.45 \pm 0.01$ |
| Percentage (%) | $3.77 \pm 5.12$ | $4.43 \pm 4.96$ | $2.38 \pm 2.86$ | $0.93 \pm 1.17$ | $0.82 \pm 1.09$ | $0.59 \pm 0.78$ | $0.25 \pm 0.32$ |

Both the theoretical and empirical results demonstrate that signature generation introduces negligible overhead. In contrast to methods that require training additional models or performing repeated optimization, our CITED framework maintains a very efficient computational profile.

# F  REPRODUCIBILITY

## F.1  THREAT MODEL

To evaluate the effectiveness of our CITED framework under MEAs, we adopt a transductive threat model inspired by GNNStealing (Shen et al., 2022). While GNNStealing focuses on an inductive setting, our work targets the more practical transductive scenario, typical in applications such as social (Tang & Liu, 2010; Zhang et al., 2022) or financial networks (Wang et al., 2021), where the graph structure is fixed and globally known, and attackers can only query subgraphs. Specifically, we consider a *Type I* attack, where the attacker queries the target model $f_T$ on a subgraph $\mathcal{G}_{\text{query}} = (\mathcal{V}_{\text{query}}, \mathcal{E}_{\text{query}})$ from the same graph used for training. The model returns outputs $\mathcal{O}^*_{\text{query}} = f_T(\mathcal{V}_{\text{query}}, \mathcal{E}_{\text{query}})$, representing either node embeddings or labels depending on the output level. Extending GNNStealing, we assess verification performance under both embedding- and label-level outputs, enabling a more comprehensive and realistic evaluation.

## F.2  IMPLEMENTATION OF THREAT MODEL

We base our approach on the assumption that, in order to obtain a surrogate model whose performance closely approximates that of the target model, the attacker is often compelled to query information near the decision boundary. Accordingly, we design a generalizable threat model. Specifically, we identify samples with the most ambiguous prediction probabilities as representative of decision boundary information. To simulate attacker queries, we construct a query set $S_{\text{query}}$, where 20% of the samples correspond to boundary information, and the remaining 80% are randomly selected.

## F.3  PACKAGES REQUIRED FOR IMPLEMENTATIONS

```
Python==3.13.3
matplotlib==3.10.1
numpy==2.2.5
scikit-learn==1.6.1
scipy==1.15.2
torch==2.7.0
torch-geometric==2.6.1
```

## F.4  ARCHITECTURE OF BACKBONE MODELS

In our experiments, we adopt five representative graph neural network (GNN) architectures: GCN (Kipf & Welling, 2016), GAT (Veličković et al., 2017), GraphSAGE (Hamilton et al., 2017b), GCNII (Chen et al., 2020), and FAGCN (Bo et al., 2021). For consistency and fair comparison, all models share a unified architecture: each consists of two graph convolutional layers followed by a linear classifier. The node embeddings used for downstream tasks and ownership verification are obtained from the output of the final convolutional layer.

We apply a dropout rate of 0.5 for all models except GAT. For GAT, we follow the standard setting with a dropout rate of 0.5 and use 8 attention heads. For GCNII, we set the initial residual connection strength $\alpha = 0.1$ and the identity mapping strength $\beta = 0.5$, consistent with the original implementation. For training the target model, we employ the Adam optimizer with a learning rate of 0.001 and a weight decay of $1 \times 10^{-5}$. The model is trained for 200 epochs using the cross-entropy loss function.

## F.5  PARAMETERS FOR CITED FRAMEWORK

Our proposed CITED framework involves two key hyperparameters for signature generation: the boundary selection ratio, set to `cited_boundary_ratio` = 0.1, and the signature node selection ratio, set to `cited_signature_ratio` = 0.2. These parameters control the proportion of nodes identified near the decision boundary and select nodes surrounding them to construct the final signature set.

After generating the signature set, we fine-tune the target model using only the original task labels. This finetuning is performed with a learning rate of 0.001 and a weight decay of 1e-5. The model is tuned for 50 epochs. Notably, the process does not involve any task-irrelevant or external data, ensuring that the original task semantics remain intact. As a result, unlike many watermarking-based approaches that may incur performance degradation, our framework typically preserves in many cases even slightly improves the original model's accuracy.

### F.6 PARAMETERS FOR BASELINE METHODS

For all baseline models, we adopt a unified training configuration to ensure fair comparison. Specifically, we use the Adam optimizer with a learning rate of 0.001 and a weight decay of 1e-5. All models are trained using the cross-entropy loss for 200 epochs.

To account for randomness, each experiment is repeated three times with different random seeds, and we report the mean and standard deviation of the results. In particular, for the target models used in defense methods, we select the model checkpoint with the highest performance among the three runs as the protected target model.

We then provide the specific hyperparameter we widely used in our experiments.

**GrOVe:** We strictly follow the experimental settings described in the original paper, and no special hyperparameter tuning is required for the defense methods.

**RandomWM:** We set `random_node_num = 10`, `random_edge_ratio = 0.3`, and `random_feat_ratio = 0.1`, which respectively control the number of randomly selected nodes, the ratio of randomly added edges, and the ratio of perturbed features.

**BackdoorWM:** We configure `backdoor_ratio = 0.05` and `backdoor_len = 20`, specifying the proportion of backdoor triggers in the graph and the length of the backdoor pattern, respectively.

**SurviveWM:** We set `survive_node_num = 10` and `survive_edge_prob = 0.5`, which define the number of watermark nodes and the probability of edge preservation among them.

### F.7 COMPUTING RESOURCES

All training, inference, and efficiency evaluations in our experiments were conducted on a high-performance computing server equipped with an Nvidia RTX 6000 Ada GPU. The system is powered by an AMD EPYC 7763 64-Core processor running at 2.45 GHz, offering a total of 128 threads. The server is configured with 1000 GB of DDR4 RAM (registered and buffered), operating at a memory speed of 3200 MT/s and manufactured by Samsung.

## G   LLM USAGE STATEMENT

Large Language Models (LLMs) were not involved in generating research ideas, designing algorithms, or conducting experiments. They were only used to improve the clarity and readability of the manuscript, including grammar and phrasing. All methodological innovations, theoretical insights, and experimental validations were solely developed and verified by the authors.

