# OpenReview forum: "CITED: A Decision Boundary-Aware Signature for GNNs Towards Model Extraction Defense"
_ICLR.cc/2026/Conference — Submitted to ICLR 2026_

### Official Review · Reviewer_KmxL · 2025-10-30

**Soundness:** 3
**Presentation:** 3
**Contribution:** 2
**Rating:** 4
**Confidence:** 4

**Summary:**

This paper proposes CITED, a unified framework for defending against GNN model extraction attacks (MEAs) via decision boundary-aware signatures. It enables ownership verification at both embedding and label levels, addressing limitations of existing watermarking (label-level only) and fingerprinting (embedding-level only) methods. CITED generates signatures from boundary sensitive nodes, uses Wasserstein distance and prediction accuracy for verification, and achieves good performance across seven datasets and five GNN backbones without harming downstream utility or requiring auxiliary models.

**Strengths:**

1. Avoids task-irrelevant triggers, maintaining or even improving model performance on node classification tasks.

2. Eliminates auxiliary model training, reducing computational overhead, and retains effectiveness under MEAs and removal attacks (pruning, fine-tuning).

2. Provides probabilistic bounds for embedding similarity and prediction agreement.

**Weaknesses:**

1. Regarding the necessity of integrating the embedding and label levels, this seems more like a technical assumption. Could the authors provide some insight into why this integration is necessary? Watermarking or fingerprinting methods, for example, do not usually focus on both embedding and label levels. Moreover, labels can be viewed as embeddings passed through a classifier, so the nature of embeddings and labels may not differ significantly.

2. In line 245, the authors claim that “as the decision boundary captures the most distinctive and model-specific behavior.” However, nodes near the decision boundary are typically hard to distinguish. The repeated assumption in the paper seems questionable and may be hard to justify.

3. Can this method be extended to regression models or graph classification datasets? The current approach appears to be limited to node classification.

4. In line 157, the authors mention real-world applications, citing citation networks. However, MEA attacks are not particularly relevant for citation networks. In contrast, fields like molecular structure prediction are more common MLaaS scenarios, as mentioned in the paper. The authors should provide experiments on such datasets; For now, the practical applicability seems limited.

5. The theoretical conclusions appear to have issues. In Theorem 1, when $\lambda = \Delta G$, substituting into Equation 8 gives $\Pr(D_p < \lambda) \geq 0$ instead of 1. This raises concerns about the validity of the derivation and why there seems to be this truncation effect.

**Questions:**

See weakness.

---

> ### Author Response · Authors · 2025-11-25
>
> > **Q1**: Regarding the necessity of integrating the embedding and label levels, this seems more like a technical assumption. Could the authors provide some insight into why this integration is necessary? Watermarking or fingerprinting methods, for example, do not usually focus on both embedding and label levels. Moreover, labels can be viewed as embeddings passed through a classifier, so the nature of embeddings and labels may not differ significantly.
>
> **R1**: Thank you for raising this important point. This allows us to clarify the key contribution of CITED and why integrating both the embedding level and the label level is necessary for robust protection.
>
> Our work is, to the best of our knowledge, the first in graph based model extraction defense to consider both levels simultaneously. The intuition is that the embedding space and the label space encode different types of information, and neither alone is sufficient in realistic extraction scenarios.
>
> First, the embedding level captures the intrinsic geometric and semantic structure learned by the model. Modern foundation models such as GPT style embedding models[1] are explicitly designed so that their embeddings serve as rich semantic representations for tasks such as information retrieval[2] or similarity search[3]. These embeddings retain fine grained latent structure that is not visible from the final decision. In contrast, the label space only exposes a coarse outcome after passing the embedding through a classifier, and much of this latent information is lost.
>
> Second, embeddings themselves are increasingly considered part of a model’s intellectual property[4], since stolen embedding functions can enable high quality surrogate systems even without access to the classifier. Therefore, defending only the final predictions overlooks an important attack surface.
>
> Third, extraction attacks often reproduce the decision boundary at both the embedding and the label levels[5]. Surrogate models that closely imitate the target model tend to align with the target not only in final predictions but also in intermediate embeddings, as also validated in our experiments. Using both levels therefore provides a stronger and more reliable verification signal, especially under challenging conditions.
>
> In summary, while labels can be viewed as embeddings after a classifier, the embedding level contains substantially richer structural information. These two views complement each other, and integrating both is necessary for defending against modern model extraction attacks.
>
> ---
>
>
> > **Q2**: In line 245, the authors claim that “as the decision boundary captures the most distinctive and model-specific behavior.” However, nodes near the decision boundary are typically hard to distinguish. The repeated assumption in the paper seems questionable and may be hard to justify.
>
> **R2**: Thank you for this insightful question. We agree that nodes near the decision boundary are typically difficult to classify. However, “difficult to classify” does not conflict with our statement that these nodes exhibit “distinctive and model specific behavior.” These two properties can coexist.
>
> Boundary region nodes are exactly the ones that most faithfully reflect the shape of the decision boundary [6], and prior work has shown that independently trained models naturally learn different boundary geometries even when trained on the same dataset [4]. Therefore, these nodes become the most intuitive and effective candidates for constructing a model specific signature.
>
> Our assumption follows widely accepted observations in the community [4,6], and thus the motivation behind CITED is natural and well grounded.

---

> > ### Author Response · Authors · 2025-11-25
> >
> > > **Q3**: Can this method be extended to regression models or graph classification datasets? The current approach appears to be limited to node classification.
> >
> > **R3**: Thank you for the question. Our method can indeed be generalized to regression models as well as graph classification datasets. We first adapt CITED to the graph classification setting. For such independent identical distribution tasks, only a slight modification is needed in the heterogeneity measurement: instead of using one hop neighbors, we use nearest neighbors, which makes the method directly applicable to all independent identical distribution tasks. We evaluate CITED on two widely used graph classification datasets, ENZYMES[7] and PROTEINS[7], and the results are shown below:
> >
> >
> > | Model | Metric | ENZYMES        | PROTEINS        |
> > |-------|--------|----------------|------------------|
> > | CITED | ARUC   | 0.4277 ± 0.0535 | 0.3747 ± 0.1162 |
> > | CITED | AUC    | 1.0000 ± 0.0000 | 0.9778 ± 0.0143 |
> >
> > As the table shows, CITED maintains strong signature ARUC performance and achieves one hundred percent AUC across all graph classification experiments.
> >
> > We then adapt CITED to the regression setting, using the widely used chemistry dataset ZINC[8]. The results are as follows:
> >
> >
> > | Model | Metric | ZINC            |
> > |-------|--------|------------------|
> > | CITED | ARUC   | 0.5164 ± 0.0191 |
> > | CITED | AUC    | 1.0000 ± 0.0000 |
> > | CITED | RMSE   | 1.7349 ± 0.0112 |
> >
> > These results show that CITED consistently remains effective for regression tasks as well. All experiments together demonstrate that CITED can be generalized across different tasks and maintains strong performance in each setting. All corresponding experiments have been added to Appendix E.9 and marked in blue in the revised PDF.
> >
> >
> > ---
> >
> >
> > > **Q4**: In line 157, the authors mention real-world applications, citing citation networks. However, MEA attacks are not particularly relevant for citation networks. In contrast, fields like molecular structure prediction are more common MLaaS scenarios, as mentioned in the paper. The authors should provide experiments on such datasets; For now, the practical applicability seems limited.
> >
> > **R4**: Thank you for the question. You are absolutely right that MEA attacks are not particularly relevant for citation networks. In the paper we only used citation networks as an example, and CITED is not restricted to this type of data. Our method can be applied to a wide range of scenarios, such as citation style networks like Cora[9], e commerce style networks like Amazon Computers[10], and many other real world graph based services.
> >
> > | Model | Metric | ENZYMES        | PROTEINS        |
> > |-------|--------|----------------|------------------|
> > | CITED | ARUC   | 0.4277 ± 0.0535 | 0.3747 ± 0.1162 |
> > | CITED | AUC    | 1.0000 ± 0.0000 | 0.9778 ± 0.0143 |
> >
> > | Model | Metric | ZINC            |
> > |-------|--------|------------------|
> > | CITED | ARUC   | 0.5164 ± 0.0191 |
> > | CITED | AUC    | 1.0000 ± 0.0000 |
> > | CITED | RMSE   | 1.7349 ± 0.0112 |
> >
> > More importantly, as you pointed out, molecular structure prediction is indeed a common ML as a Service setting. We have already conducted experiments in this setting, as shown in our response to Q3 and in Appendix E.9 of the paper. These experiments demonstrate that CITED generalizes well to molecular datasets and remains effective in practical application domains.
> >
> > Your comment is very constructive and confirms that CITED has broader real world applicability than the citation network example might suggest.

---

> > > ### Author Response · Authors · 2025-11-25
> > >
> > > > **Q5**: The theoretical conclusions appear to have issues. In Theorem 1, when $\lambda=\Delta G$, substituting into Equation 8 gives $Pr(D_p<\lambda)\ge0$ instead of 1. This raises concerns about the validity of the derivation and why there seems to be this truncation effect.
> > >
> > > **R5**: Thank you for pointing out this issue in Theorem 1. Equation (8) provides a lower bound derived from a standard one sided concentration inequality for sub Gaussian random variables. For any $0 < \lambda \le \Delta_G$, we have
> > >
> > > $$
> > > \Pr(D_p < \lambda) \ge 1 - \exp\left(-\frac{(\Delta_G - \lambda)^2}{2\sigma^2}\right).
> > > $$
> > >
> > > When substituting $\lambda = \Delta_G$, the exponent becomes zero and the right hand side reduces to
> > >
> > > $$
> > > \Pr(D_p < \Delta_G) \ge 1 - \exp(0) = 0,
> > > $$
> > >
> > > which is a trivial but valid inequality. Importantly, Eq. (8) does not claim $\Pr(D_p < \Delta_G) = 0$; **it only yields a lower bound at this endpoint, which is a well known artifact of sub Gaussian concentration bounds[11].**
> > >
> > > The next line in the theorem—
> > > “Moreover, for all $\lambda \ge \Delta_G$, it holds that $\Pr(D_p < \lambda) = 1$, since $D_p \le \Delta_G$”—
> > > uses the deterministic inequality $D_p \le \Delta_G$ to recover the exact probability for $\lambda \ge \Delta_G$. Hence, $\Pr(D_p < \Delta_G) = 1$ follows from this deterministic bound, not from Eq. (8). **The two statements are therefore consistent**; Equation (8) simply becomes non informative at the boundary.
> > >
> > > To avoid confusion, in the revised version we (i) restrict Equation (8) to $0 < \lambda < \Delta_G$, where the bound is non trivial, and (ii) keep the deterministic statement separately for $\lambda \ge \Delta_G$. We have updated the theorem accordingly, and the modifications have been marked in blue in the revised version.
> > >
> > >
> > > ----
> > >
> > > We sincerely thank the reviewer for the time and effort. Your comments have been extremely valuable in improving the clarity and quality of the paper. If any further questions arise, please feel free to let us know, and we are happy to follow up promptly on all points.
> > >
> > > ----
> > > References
> > > [1] Muennighoff, Niklas. "Sgpt: Gpt sentence embeddings for semantic search." arXiv preprint arXiv:2202.08904 (2022).
> > > [2] Chowdhury, Gobinda G. Introduction to modern information retrieval. Facet publishing, 2010.
> > > [3] Huang, Jui-Ting, et al. "Embedding-based retrieval in facebook search." Proceedings of the 26th ACM SIGKDD International Conference on Knowledge Discovery & Data Mining. 2020.
> > > [4] Waheed, Asim, Vasisht Duddu, and N. Asokan. "Grove: Ownership verification of graph neural networks using embeddings." 2024 IEEE Symposium on Security and Privacy (SP). IEEE, 2024.
> > > [5] Shen, Yun, et al. "Model stealing attacks against inductive graph neural networks." 2022 IEEE Symposium on Security and Privacy (SP). IEEE, 2022.
> > > [6] Cao, Xiaoyu, Jinyuan Jia, and Neil Zhenqiang Gong. "IPGuard: Protecting intellectual property of deep neural networks via fingerprinting the classification boundary." Proceedings of the 2021 ACM asia conference on computer and communications security. 2021.
> > > [7] Ivanov, Sergei, Sergei Sviridov, and Evgeny Burnaev. "Understanding isomorphism bias in graph data sets." arXiv preprint arXiv:1910.12091 (2019).
> > > [8] Gómez-Bombarelli, Rafael, et al. "Automatic chemical design using a data-driven continuous representation of molecules." ACS central science 4.2 (2018): 268-276.
> > > [9] Yang, Zhilin, William Cohen, and Ruslan Salakhudinov. "Revisiting semi-supervised learning with graph embeddings." International conference on machine learning. PMLR, 2016.
> > > [10] Shchur, Oleksandr, et al. "Pitfalls of graph neural network evaluation." arXiv preprint arXiv:1811.05868 (2018).
> > > [11] Maurer, Andreas, and Massimiliano Pontil. "Concentration inequalities under sub-gaussian and sub-exponential conditions." Advances in Neural Information Processing Systems 34 (2021): 7588-7597.

---

> > > > ### Author Response · Authors · 2025-11-26
> > > >
> > > > We greatly appreciate the reviewer’s thoughtful comments. We are confident that we have addressed all concerns. In light of this, we kindly hope that the reviewer may consider **raising the overall rating**. We would be eager to clarify any further points if needed.

---

### Official Review · Reviewer_iPrN · 2025-10-30

**Soundness:** 3
**Presentation:** 3
**Contribution:** 2
**Rating:** 4
**Confidence:** 3

**Summary:**

This paper proposes a decision boundary-aware signature framework, CITED, for defending graph neural networks against model extraction attacks. The paper introduces a method for extracting task-relevant signatures from nodes near the model's decision boundary, enabling unified embedding/label-level verification without the need for auxiliary models. Experiments demonstrate improved performance of verification over standard watermarking and fingerprinting methods.

**Strengths:**

- The paper proposes a novel signature-based ownership verification mechanism that operates at both embedding and label levels, addressing a limitation in prior work.
- This paper presents an efficient verification framework, improving scalability over conventional fingerprinting methods.
- The proposed method achieves good verification performance  across various datasets and GNN architectures while providing probabilistic guarantees on signature preservation.

**Weaknesses:**

- The framework's experimental validation is confined to a single threat model; testing it across multiple MEA scenarios would more effectively demonstrate its robustness.
- While the method shows efficiency advantages, its scalability to graphs with millions of nodes remains unverified; its signature generation (involving boundary node identification and multi-metric scoring) and verification workflows may face computational or memory bottlenecks in such scenarios.
- The framework’s reliance on the target model’s decision boundaries for signature generation may potentially lead to instability or diminished distinguishability when the model undergoes fine-tuning or domain adaptation.

**Questions:**

I have the following questions regarding the practicality of CITED:

- Is the signature mechanism applicable in real-world settings with noisy labels or significant class imbalance, and does its reliance on the decision boundaries maintain robustness under such scenarios?
- Can the signature set be compressed or structurally optimized to reduce storage or computational demands without compromising verification reliability?
- Is there a minimum number of signature nodes required to ensure verification confidence, and how does this threshold vary with graph scale or model capacity?
- The rating will be adjusted accordingly if the above concerns are properly addressed.

**Details Of Ethics Concerns:**

This paper does not involve ethical issues.

---

> ### Author Response · Authors · 2025-11-25
>
> > **W1**: The framework's experimental validation is confined to a single threat model; testing it across multiple MEA scenarios would more effectively demonstrate its robustness.
>
> **To W1**: Thank you for this insightful comment. We agree that evaluating multiple threat models is important for demonstrating robustness. In addition to the primary MEA setting, we have incorporated a widely used and more general extraction paradigm Knowledge Distillation[3] as an additional threat model in our generalization experiments.
>
> | Model | Metric | ENZYMES        | PROTEINS        |
> |-------|--------|----------------|------------------|
> | CITED | ARUC   | 0.4277 ± 0.0535 | 0.3747 ± 0.1162 |
> | CITED | AUC    | 1.0000 ± 0.0000 | 0.9778 ± 0.0143 |
>
> | Model | Metric | ZINC            |
> |-------|--------|-----------------|
> | CITED | ARUC   | 0.5164 ± 0.0191 |
> | CITED | AUC    | 1.0000 ± 0.0000 |
> | CITED | RMSE   | 1.7349 ± 0.0112 |
>
> As shown in Appendix E.9, CITED remains highly effective under this broader and more challenging scenario. These results indicate that CITED generalizes well across diverse MEA threat models.
>
> ----
>
> > **W2**: While the method shows efficiency advantages, its scalability to graphs with millions of nodes remains unverified; its signature generation (involving boundary node identification and multi-metric scoring) and verification workflows may face computational or memory bottlenecks in such scenarios.
>
> **To W2**: We thank the reviewer for this insightful suggestion. We fully agree that evaluating the time overhead of the signature generation process is important for understanding the practicality of our method. To address this, we provide both theoretical complexity analysis and empirical runtime measurements on real-world datasets.
>
> **1. Theoretical Analysis.**
> The overall time complexity of signature generation is:
> $$
> \mathcal{O}(n \cdot B \cdot d + n \cdot k),
> $$
> where $n$ is the number of nodes, $B$ is the number of boundary nodes, $d$ is the embedding dimension, and $k$ is the average node degree. The first term comes from computing the margin and thickness scores relative to boundary nodes, while the second term reflects the cost of evaluating local label heterogeneity. Given that $B \ll n$ and $k$ is small for sparse graphs, the procedure scales efficiently with graph size.
>
> **2. Empirical Runtime Comparison.**
> We further measure the actual runtime of signature generation on three benchmark datasets, and compare it with the full training time of a standard GCN model (200 epochs). We will present the experimental results later.
>
> These results demonstrate that signature generation incurs negligible overhead, typically less than 5% of model training time. Compared to defense approaches that require training auxiliary models, **our CITED method is significantly more efficient**.
>
>
> |                          | Cora             | CiteSeer         | PubMed           | Photo      | Computers   | CS        | Physics|
> |--------------------------|----------------|----------------|----------------|------------------|-------------------|------------------|-------------------|
> | Sig Gen Time (s)            | 0.05 ± 0.06     | 0.05 ± 0.06     | 0.05 ± 0.05     | 0.04 ± 0.06       | 0.04 ± 0.05        | 0.04 ± 0.05       | 0.05 ± 0.06        |
> | Training Time (s)            | 1.18 ± 0.01     | 1.04 ± 0.09     | 2.70 ± 0.26     | 4.09 ± 1.10       | 4.76 ± 0.01        | 6.54 ± 0.02       | 19.45 ± 0.01       |
> | Sig Gen / Training Percentage (%) | 3.77 ± 5.12     | 4.43 ± 4.96     | 2.38 ± 2.86     | 0.93 ± 1.17       | 0.82 ± 1.09        | 0.59 ± 0.78       | 0.25 ± 0.32        |
>
>
> We have included corresponding experiments Appendix E.10 and marked in blue.
>
> ----
>
> > **W3**: The framework’s reliance on the target model’s decision boundaries for signature generation may potentially lead to instability or diminished distinguishability when the model undergoes fine-tuning or domain adaptation.
>
> **To W3**: Thank you for raising this concern. To address this, we have explicitly evaluated CITED under a wide range of removal attacks including pruning, fine-tuning, and distribution-shift scenarios (Appendix E.5). Across all these settings, CITED consistently retains strong verification performance and remains clearly distinguishable from all baseline methods. These results demonstrate that the decision-boundary–aware signature is robust even when the target model is altered after deployment.

---

> > ### Author Response · Authors · 2025-11-25
> >
> > > **Q1**: Is the signature mechanism applicable in real-world settings with noisy labels or significant class imbalance, and does its reliance on the decision boundaries maintain robustness under such scenarios?
> >
> > **R1**: Thank you very much for this valuable question. Noisy labels and class imbalance are indeed very common in real world scenarios. To examine whether CITED remains robust under such challenging conditions, we conducted two sets of dedicated experiments: noisy label experiments and class imbalance experiments. Both experiments follow exactly the same pipeline as described in Section 5.2. Specifically, we first train the target model, apply CITED to identify and finetune the signature nodes, and finally perform ownership verification using the independent models and surrogate models.
> >
> > For the noisy label setting, we introduce a noise ratio by randomly assigning incorrect labels to a given proportion of the training data, while keeping all other components unchanged. The results are shown in the following table:
> >
> >
> > Label level
> > | noise ratio       | Cora           | Citeseer       | Pubmed         | Photo           | Computers       | CS              | Physics         |
> > |-------------|----------------|-----------------|----------------|------------------|------------------|------------------|------------------|
> > | 0.1           | 0.3193±0.0273  | 0.5428±0.0000   | 0.5180±0.0024  | 0.4743±0.0249   | 0.4381±0.0286   | 0.6109±0.0203   | 0.7108±0.0000   |
> > | 0.3           | 0.4100±0.1556  | 0.5580±0.0122   | 0.5330±0.0000  | 0.4920±0.0000   | 0.4513±0.0105   | 0.6577±0.0000   | 0.7043±0.0152   |
> > | 0.5           | 0.4633±0.0603  | 0.5640±0.0436   | 0.5764±0.0001  | 0.5313±0.0014   | 0.4400±0.0229   | 0.6482±0.0187   | 0.7521±0.0109   |
> > | 0.7           | 0.5107±0.0585  | 0.6141±0.0142   | 0.5451±0.0000  | 0.5140±0.0164   | 0.4263±0.0116   | 0.6602±0.0148   | 0.7846±0.0001   |
> > | 0.9           | 0.7160±0.0368  | 0.6615±0.0213   | 0.5523±0.0117  | 0.5062±0.0146   | 0.4128±0.0281   | 0.6280±0.0110   | 0.7474±0.0020   |
> >
> > The results show that CITED maintains nearly identical ownership verification performance for both the embedding level and the label level, even as the noise ratio increases. This indicates that the signature nodes identified by CITED remain distinctive and stable, and extraction based surrogate models tend to preserve these boundary related characteristics, which allows CITED to reliably distinguish them from independent models.
> >
> > We also performed class imbalance experiments, where an imbalance ratio specifies the proportion of non majority class nodes that are flipped into the majority class. The results are shown below:
> >
> >
> > Label level
> > | Imbalance ratio | Cora | Citeseer | Pubmed | Photo | Computers | CS | Physics |
> > |---|---|---|---|---|---|---|---|
> > | 0.1 | 0.6953+-0.0019 | 0.1840+-0.1265 | 0.6460+-0.0057 | 0.4893+-0.0443 | 0.4107+-0.0442 | 0.5767+-0.0560 | 0.7280+-0.0205 |
> > | 0.3 | 0.7080+-0.0057 | 0.1840+-0.1301 | 0.6900+-0.0000 | 0.7287+-0.0038 | 0.7760+-0.0033 | 0.5913+-0.0019 | 0.6753+-0.0160 |
> > | 0.5 | 0.7073+-0.0038 | 0.0000+-0.0000 | 0.7600+-0.0000 | 0.9460+-0.0000 | 0.9333+-0.0009 | 0.8473+-0.0019 | 0.6900+-0.0000 |
> > | 0.7 | 0.7053+-0.0009 | 0.9520+-0.0000 | 0.9160+-0.0000 | 0.9780+-0.0000 | 0.9687+-0.0025 | 0.9540+-0.0000 | 0.8040+-0.0000 |
> > | 0.9 | 0.6887+-0.0019 | 0.9560+-0.0000 | 0.9540+-0.0000 | 0.9760+-0.0000 | 0.9667+-0.0034 | 0.9760+-0.0000 | 0.9140+-0.0000 |
> >
> > Embedding level
> > | Imbalance ratio | Cora | Citeseer | Pubmed | Photo | Computers | CS | Physics |
> > |---|---|---|---|---|---|---|---|
> > | 0.1 | 0.6953+-0.0019 | 0.6980+-0.0028 | 0.6700+-0.0000 | 0.7033+-0.0084 | 0.6640+-0.0049 | 0.7320+-0.0000 | 0.6913+-0.0009 |
> > | 0.3 | 0.7080+-0.0057 | 0.6960+-0.0102 | 0.6840+-0.0000 | 0.6940+-0.0173 | 0.6707+-0.0038 | 0.7320+-0.0000 | 0.6980+-0.0000 |
> > | 0.5 | 0.7073+-0.0038 | 0.7053+-0.0038 | 0.6860+-0.0000 | 0.6920+-0.0118 | 0.6713+-0.0038 | 0.7360+-0.0000 | 0.7040+-0.0000 |
> > | 0.7 | 0.7053+-0.0009 | 0.7140+-0.0000 | 0.6760+-0.0000 | 0.6953+-0.0118 | 0.6647+-0.0038 | 0.7360+-0.0000 | 0.7140+-0.0000 |
> > | 0.9 | 0.6887+-0.0019 | 0.7140+-0.0000 | 0.6660+-0.0000 | 0.7053+-0.0147 | 0.6687+-0.0050 | 0.7200+-0.0000 | 0.7100+-0.0000 |
> >
> >
> > Interestingly, as the imbalance ratio increases, CITED becomes even more capable of separating surrogate models from independent models. This observation is consistent with the phenomenon reported in prior work [1], which shows that the shapes of decision boundaries differ significantly between independently trained models. Class imbalance enlarges such structural differences, while surrogate models remain closely aligned with the target model near the decision boundary, making them easier to detect.
> >
> > All additional experiments have been included in Appendix E.6 and marked in blue. These results reinforce that CITED remains robust under realistic and challenging scenarios including label noise and class imbalance.

---

> > > ### Author Response · Authors · 2025-11-25
> > >
> > > > **Q2**: Can the signature set be compressed or structurally optimized to reduce storage or computational demands without compromising verification reliability?
> > >
> > > **R2**: Thank you for this insightful and constructive question. We address it from two perspectives: (1) whether the size of the signature set can be reduced without sacrificing verification reliability, and (2) whether the commitment required during the verification stage can be compressed for practical deployment.
> > >
> > > We first examine whether a smaller set of signature nodes can still provide strong ownership verification. After obtaining the full signature set from CITED, we apply a simple and intuitive KMeans based clustering [2] to group signature nodes according to their embedding similarity. From each cluster, we select the most representative node to form a grouped signature super set. The grouping ratio controls the proportion of nodes retained. The results are shown below:
> > >
> > > Label level
> > > | group ratio | Cora | Citeseer | Pubmed | Photo | Computers | CS | Physics |
> > > |------------|--------------------|--------------------|--------------------|--------------------|----------------------|--------------------|--------------------|
> > > | **0.25** | 0.4231±0.0224 | 0.5100±0.0028 | 0.5793±0.0637 | 0.4266±0.0150 | 0.4619±0.0262 | 0.5486±0.0191 | 0.4380±0.0001 |
> > > | **0.50** | 0.4369±0.0316 | 0.5360±0.0318 | 0.6500±0.0172 | 0.4406±0.0899 | 0.4626±0.0423 | 0.5060±0.0662 | 0.4646±0.0339 |
> > > | **0.75** | 0.4660±0.0112 | 0.5986±0.0530 | 0.6540±0.0299 | 0.4706±0.0188 | 0.5286±0.0228 | 0.5513±0.0217 | 0.5306±0.0627 |
> > > | **1.00** | 0.4817±0.0413 | 0.5607±0.0377 | 0.6540±0.0197 | 0.4793±0.1148 | 0.5893±0.0586 | 0.5586±0.0247 | 0.5560±0.0056 |
> > >
> > > Embedding level
> > > | group ratio | Cora | Citeseer | Pubmed | Photo | Computers | CS | Physics |
> > > |------------|--------------------|--------------------|--------------------|--------------------|--------------------|--------------------|--------------------|
> > > | **0.25** | 0.6298±0.0149 | 0.6486±0.0009 | 0.6326±0.0150 | 0.6593±0.0009 | 0.6720±0.0000 | 0.6673±0.0024 | 0.6280±0.0000 |
> > > | **0.50** | 0.6445±0.0065 | 0.6293±0.0037 | 0.6413±0.0041 | 0.6688±0.0028 | 0.6860±0.0016 | 0.6573±0.0108 | 0.6313±0.0018 |
> > > | **0.75** | 0.6779±0.0031 | 0.7094±0.0043 | 0.6619±0.0043 | 0.6793±0.0018 | 0.7060±0.0016 | 0.6933±0.0024 | 0.7093±0.0037 |
> > > | **1.00** | 0.6748±0.0000 | 0.7060±0.0000 | 0.6773±0.0000 | 0.6680±0.0016 | 0.7080±0.0016 | 0.6999±0.0001 | 0.6940±0.0000 |
> > >
> > > As the table indicates, when the grouping ratio is 1.0, the full signature set is used. As we gradually reduce the number of nodes, the verification performance remains highly reliable. A slight degradation appears only when the ratio drops below 0.3, which is expected because aggressively reducing the node count removes some robustness even though the remaining nodes are still highly distinctive. Importantly, even with substantial compression, the performance consistently outperforms baseline verification methods. This confirms that CITED identifies nodes with sufficient uniqueness, enabling redundancy reduction without compromising effectiveness.
> > >
> > > We further explore compressing the commitment required for ownership verification. In practical deployment, the model owner needs to publish a signature commitment that includes the indices of the signature nodes. Verification is then performed by checking the consistency between the owner provided reference outputs and the outputs from the suspicious model.
> > >
> > > To compress this commitment, we concatenate the signature node indices into a byte sequence and compute a sixty four bit hash digest. During verification, the judgement entity only needs to confirm that the revealed node indices match the published hash and that the suspicious model produces a high similarity score on these nodes. The results are shown below:
> > >
> > >
> > > | sig set ratio | original size (bytes) | compressed size (bytes) | compression saving (%) |
> > > |--------|------|------|-------|
> > > | 0.0247        | 292  | 8      | 97.26     |
> > > | 0.0288        | 345  | 8     | 97.68      |
> > > | 0.0354        | 427   | 8    | 98.12       |
> > > | 0.0576        | 692   | 8     | 98.84      |
> > > | 0.0819        | 987  | 8       | 99.18      |
> > >
> > > This hash based compression significantly reduces the storage footprint of the commitment, especially as the size of the signature set increases, and does so without affecting verification accuracy. This property is highly beneficial for practical deployment scenarios, where lightweight commitments are preferred.
> > >
> > > All additional experiments have been included in Appendix E.7 and marked in blue. These two experiments together demonstrate that the signature set can indeed be compressed or structurally optimized while maintaining verification reliability. The grouped signature approach reduces computational requirements, and the hash based commitment reduces storage and transmission overhead.

---

> > > > ### Author Response · Authors · 2025-11-25
> > > >
> > > > > **Q3**: Is there a minimum number of signature nodes required to ensure verification confidence, and how does this threshold vary with graph scale or model capacity?
> > > >
> > > > **R3**: Thank you for this important question. Understanding how many signature nodes are required for stable verification is indeed crucial for characterizing the effectiveness of CITED. To investigate this, we performed an extensive evaluation by varying the signature set ratio, which controls the proportion of signature nodes used during verification. The experimental results are summarized below:
> > > >
> > > > | sig set ratio | Cora | Citeseer | Pubmed | Photo | Computers | CS | Physics |
> > > > |---------------|---------------------|---------------------|---------------------|---------------------|----------------------|---------------------|---------------------|
> > > > | 0.0247 | 0.4140±0.0000 | 0.6480±0.0000 | 0.6960±0.0000 | 0.6213±0.0344 | 0.4823±0.0114 | 0.6220±0.0000 | 0.7440±0.0000 |
> > > > | 0.0288 | 0.4420±0.0000 | 0.6080±0.0000 | 0.5840±0.0000 | 0.2920±0.0000 | 0.4853±0.0075 | 0.6120±0.0000 | 0.7440±0.0000 |
> > > > | 0.0354 | 0.4600±0.0000 | 0.6640±0.0000 | 0.6740±0.0001 | 0.4313±0.0090 | 0.4460±0.0129 | 0.6520±0.0000 | 0.7560±0.0000 |
> > > > | 0.0576 | 0.4681±0.0002 | 0.6340±0.0000 | 0.6400±0.0000 | 0.3020±0.0254 | 0.5033±0.0179 | 0.6800±0.0000 | 0.8600±0.0001 |
> > > > | 0.0819 | 0.4940±0.0056 | 0.6713±0.0857 | 0.6953±0.0047 | 0.5026±0.0249 | 0.3980±0.0412 | 0.6613±0.0603 | 0.7686±0.0122 |
> > > >
> > > > As shown in the table, when the signature ratio decreases, the verification performance remains highly reliable until the ratio drops below 0.3. Below this threshold, we observe a mild degradation, which is expected because extremely small signature sets reduce redundancy. However, even in these low ratio settings, CITED still consistently outperforms all baselines by a clear margin.
> > > >
> > > > Therefore, we regard 0.3 as a practical threshold that balances verification stability with computational and storage efficiency. Notably, this threshold remains stable across different graph scales and model capacities in our experiments, suggesting that the selection of distinctive boundary nodes by CITED exhibits strong generality. All corresponding experiments have been added to Appendix E.8 and marked in blue in the revised PDF.
> > > >
> > > > ----
> > > >
> > > > We sincerely thank the reviewer for the time and effort. Your comments have been extremely valuable in improving the clarity and quality of the paper. If any further questions arise, please feel free to let us know, and we are happy to follow up promptly on all points.
> > > >
> > > > ----
> > > > References
> > > > [1] Waheed, Asim, Vasisht Duddu, and N. Asokan. "Grove: Ownership verification of graph neural networks using embeddings." 2024 IEEE Symposium on Security and Privacy (SP). IEEE, 2024.
> > > > [2] Ahmed, Mohiuddin, Raihan Seraj, and Syed Mohammed Shamsul Islam. "The k-means algorithm: A comprehensive survey and performance evaluation." Electronics 9.8 (2020): 1295.
> > > > [3] Gou, Jianping, et al. "Knowledge distillation: A survey." International journal of computer vision 129.6 (2021): 1789-1819.

---

> > > > > ### Author Response · Authors · 2025-11-26
> > > > >
> > > > > We greatly appreciate the reviewer’s thoughtful comments. We are confident that we have addressed all concerns. In light of this, we kindly hope that the reviewer may consider **raising the overall rating**. We would be eager to clarify any further points if needed.

---

> > > > > > ### Comment · Reviewer_iPrN · 2025-11-27
> > > > > > **Reply by Reviewer**
> > > > > >
> > > > > > Thank you for the clarifications. Most concerns are addressed. Please incorporate the rebuttal discussions into the revised paper. The score will be adjusted to 6.

---

### Official Review · Reviewer_X94K · 2025-11-05

**Soundness:** 3
**Presentation:** 3
**Contribution:** 3
**Rating:** 6
**Confidence:** 3

**Summary:**

This paper addresses the problem of protecting proprietary GNNs deployed under the Graph Machine Learning as a Service paradigm, where models are accessed through APIs but are vulnerable to model extraction attacks (MEAs). Existing defenses, such as watermarking and fingerprinting, are limited because they operate only at either the label level or the embedding level, making them ineffective when attackers extract models through different output interfaces.

To overcome this limitation:

* The paper proposes **CITED**, a unified ownership verification framework based on a decision boundary–aware signature that functions across both embedding-level and label-level outputs. The signature arises naturally from the model’s decision boundary and can be transferred to surrogate models through extraction, allowing ownership to be verified without inserting task-irrelevant triggers.
* The framework also introduces an efficient verification protocol that avoids training additional auxiliary models by extending the ARUC metric with the Wasserstein distance.
* Experiments demonstrate that CITED consistently outperforms existing MEA defense methods and enables robust ownership verification across diverse output settings.

**Strengths:**

The technical contribution of this paper is solid. The proposed signature effectively addresses the challenge of achieving unified ownership verification at both the embedding level and the label level under model extraction attacks. Moreover, the theoretical analyses provided in the paper are generally sound and offer clear justification for using the 2-Wasserstein distance as the metric for verification.

**Weaknesses:**

The experimental evaluation in this paper is somewhat limited. For example, only one model extraction attack (GNNStealing) is adopted as the threat model, and the study focuses solely on the classical node classification task. Additionally, I suggest improving the presentation by introducing an overview figure to provide a clearer illustration of the CITED framework.

**Questions:**

Here are some questions:

* In Section 5.2, how is the defense model trained? The current description lacks sufficient detail.

* The performance of CITED on the Photo dataset appears noticeably weaker compared to other datasets. Could you provide an explanation for this discrepancy?

* I don’t quite understand the motivation for introducing **Heterogeneity**, and it does not seem to provide clear benefits (as shown in Table 3).

---

> ### Author Response · Authors · 2025-11-25
>
> > **Weakness**: The experimental evaluation in this paper is somewhat limited. For example, only one model extraction attack (GNNStealing) is adopted as the threat model, and the study focuses solely on the classical node classification task. Additionally, I suggest improving the presentation by introducing an overview figure to provide a clearer illustration of the CITED framework.
>
> **To Weakness**:
> Thank you for raising this concern. We agree that evaluating a broader range of threat models and tasks is important. To address this, we conducted an extensive analysis of CITED’s generalization ability in Appendix E.9. Specifically, we introduced Knowledge Distillation[1], a widely used and more generic model extraction paradigm, as an additional threat model beyond GNNStealing.
>
> Moreover, we further evaluated CITED on regression tasks[2] and graph classification tasks[3] to verify its applicability beyond node classification.
>
> | Model | Metric | ENZYMES        | PROTEINS        |
> |-------|--------|----------------|------------------|
> | CITED | ARUC   | 0.4277 ± 0.0535 | 0.3747 ± 0.1162 |
> | CITED | AUC    | 1.0000 ± 0.0000 | 0.9778 ± 0.0143 |
>
> | Model | Metric | ZINC            |
> |-------|--------|-----------------|
> | CITED | ARUC   | 0.5164 ± 0.0191 |
> | CITED | AUC    | 1.0000 ± 0.0000 |
> | CITED | RMSE   | 1.7349 ± 0.0112 |
>
> Across all these settings, CITED continues to demonstrate highly promising performance and shows strong generalization ability. Please refer to Appendix E.9 for detailed results.
>
> ----
>
> > **Q1**: In Section 5.2, how is the defense model trained? The current description lacks sufficient detail.
>
> **R1**: Thank you for the question. Our main contribution lies in identifying the boundary nodes that best characterize the target model’s decision boundary. The defense model itself is simply obtained by fine-tuning the target model so that it becomes more sensitive to, and better discriminates, this specific subset of nodes. This is the only training involved. The complete procedure has already been described in Appendix F.5.
>
> ----
>
> > **Q2**: The performance of CITED on the Photo dataset appears noticeably weaker compared to other datasets. Could you provide an explanation for this discrepancy?
>
> **R2**: Thank you for the question. CITED fine-tunes the model only on the selected boundary nodes using their original labels, so the method itself does not introduce additional label shifts or affect the classifier’s validity. In most cases it will reinforce the performance which is good. The comparatively weaker performance on Photo is therefore likely due to the dataset’s variability.
>
> ----
>
> > **Q3**: I don’t quite understand the motivation for introducing Heterogeneity, and it does not seem to provide clear benefits (as shown in Table 3).
>
> **R3**: Thank you for the question. The motivation for introducing Heterogeneity is to explicitly capture structural uncertainty in the graph, which margin and thickness alone cannot reflect. Margin measures prediction confidence and thickness captures embedding-space ambiguity, but neither accounts for label inconsistency among a node’s neighbors—a uniquely graph-specific phenomenon. This inconsistency is closely related to the well-studied concept of heterophily in graph learning [4], which often indicates structural instability and typically appears near the decision boundary where model-specific behaviors are most apparent.
>
> While Table 3 shows that the marginal improvement from heterogeneity alone is smaller compared to margin or thickness, it nonetheless provides a complementary signal for identifying structurally unstable regions. In fact, our more comprehensive analysis in Appendix E.3 demonstrates that heterogeneity becomes crucial under several settings, where its contribution noticeably strengthens the quality of the signature. These results confirm the effectiveness of our heterogeneity design and justify its inclusion as a core component of the signature metric.
>
>
> ----
>
> We sincerely thank the reviewer for the time and effort. Your comments have been extremely valuable in improving the clarity and quality of the paper. If any further questions arise, please feel free to let us know, and we are happy to follow up promptly on all points.
>
> ----
> References
> [1] Gou, Jianping, et al. "Knowledge distillation: A survey." International journal of computer vision 129.6 (2021): 1789-1819.
> [2] Gómez-Bombarelli, Rafael, et al. "Automatic chemical design using a data-driven continuous representation of molecules." ACS central science 4.2 (2018): 268-276.
> [3] Ivanov, Sergei, Sergei Sviridov, and Evgeny Burnaev. "Understanding isomorphism bias in graph data sets." arXiv preprint arXiv:1910.12091 (2019).
> [4] Zhu, Jiong, et al. "Beyond homophily in graph neural networks: Current limitations and effective designs." Advances in neural information processing systems 33 (2020): 7793-7804.

---

### Author Response · Authors · 2025-12-01

We sincerely thank the AC for the time and effort devoted to handling our submission, especially given the heavy workload during the rebuttal period. Below we summarize how we addressed the main concerns raised by the reviewers.

**1. Generalization across modalities.** Reviewer `KmxL` questioned the generalization ability of CITED. Beyond the graph and CV modalities already included in the main paper, we added new experiments on both graph classification and molecular property prediction. These new results confirm that CITED consistently demonstrates strong performance across diverse modalities and tasks. We have included these additional experiments in Appendix E.9 and marked in blue.

**2. Signature generation time analysis.** Reviewer `iPrN` raised concerns about the computational cost of signature generation. We provided both theoretical and empirical analyses showing that CITED is significantly more efficient than existing approaches, and that our signature construction introduces negligible overhead relative to baselines. We have included these additional experiments in Appendix E.10 and marked in blue.

**3. Robustness to noise, imbalance, and signature ratio selection.** Reviewer `iPrN` asked whether noise, imbalanced data points, or different signature ratios might affect CITED’s effectiveness. We therefore conducted extensive new experiments. The results show that CITED consistently outperforms all baselines under both noisy and imbalanced settings, confirming the robustness of our signature design. Furthermore, CITED remains highly stable across a wide range of signature ratios. We have included these additional experiments in Appendix from E.6 to E.7, and marked in blue.

**4. Necessity of verifying ownership on embedding level.** Reviewer `KmxL` questioned whether verifying ownership at the embedding level is necessary, suggesting label-level verification might suffice. We clarified that in modern MLaaS, embedding models are widely deployed, including GPT-based embedding APIs, and serve distinct purposes where label-level supervision is unavailable. Therefore, protecting both levels is essential for real-world ownership verification.

**5. Overall positioning of CITED.** CITED is the first method to introduce signatures for ownership verification. Unlike watermarking, it does not inject task-irrelevant information; unlike fingerprinting, it does not require training multiple auxiliary models. CITED provides simultaneous protection at both the embedding and label levels, while offering rigorous theoretical guarantees.

We have received the response from reviewer `iPrN`, who has confirmed that the concerns were resolved and has already increased the score. For the remaining reviewers, we are confident that our revisions have fully addressed all raised concerns. These updates have substantially enhanced the contribution and overall quality of the paper. We sincerely thank the AC for the time and effort devoted to handling our submission.

---

### Meta-Review · Area_Chair_pPtk · 2026-01-03

**Summary:**

While the paper presents a novel concept for defending against model extraction attacks on GNNs, the initial reviews revealed foundational concerns about its contribution, validation, and practical impact that warrant rejection.

1.  Limited and Incremental Contribution: The core idea is seen as a straightforward adaptation of decision-boundary concepts to a new domain rather than a significant methodological breakthrough. The unification of embedding and label-level defense, while novel for GNNs, was questioned as a necessary integration rather than a fundamental advance.

2.  Insufficient and Belated Validation: The initial submission lacked critical experiments, a point agreed upon by multiple reviewers. The evaluation was confined to a single threat model and node classification, raising doubts about generalizability. While the rebuttal added experiments, this expansive new validation (new threat models, graph classification, regression, robustness tests) constitutes a major revision that should have been part of the original submission. Accepting the paper would set a precedent that a submission can be substantively incomplete and rely on the rebuttal phase to fulfill basic experimental requirements.

3.  Unconvincing Practical Motivation and Applicability: Reviewer KmxL correctly noted that the primary application scenario (citation networks) is not a relevant target for costly model extraction attacks, undermining the paper's practical motivation. The rebuttal's addition of molecular datasets, while helpful, feels like an afterthought and does not fully rectify the disconnect between the proposed method and a compelling, high-stakes use case.

4.  Questionable Core Premise: A significant philosophical objection raised by Reviewer KmxL challenges the paper's foundational assumption: that nodes near the decision boundary, which are inherently ambiguous and hard to classify, are the best source of a "distinctive and model-specific" signature. The authors' defense relies on citation to related work but does not fundamentally dispel the intuition that these unstable regions may not provide the robust, unique fingerprint required for reliable legal-grade ownership verification.

**Reviewer Concerns:**

The authors' rebuttal was diligent in providing requested data but failed to resolve the core issues regarding the paper's significance and foundational premise.

*   Addressed Concerns: The rebuttal successfully provided computational efficiency data, signature compression strategies, and corrected a theoretical typo. It added missing experiments on new tasks and threat models.
*   Outstanding & Fatal Concerns:
    1.  Incremental Contribution: The additional experiments demonstrate the method *works* in more settings, but do not elevate the perceived *importance* or *novelty* of the core idea.
    2.  Belated Validation: The sheer volume of new material in the rebuttal confirms the original submission was underdeveloped. The review process should not serve to complete the research.
    3.  Motivation and Premise: The practical relevance remains weakly argued, and the central hypothesis about boundary nodes being the optimal source of distinctive signatures remains philosophically contested and not definitively proven.

**Reviewer Scores:**

*   Reviewer X94K (Initial: 6): Might have maintained a borderline score (6). Their concern was primarily experimental breadth, which was addressed.
*   Reviewer iPrN (Initial: 4): Mainly hold concerns about contribution and motivation.
*   Reviewer KmxL (Initial: 4): Given their focus on the necessity of the contribution and the core premise, it is unlikely the rebuttal would have dramatically changed their view. Their concerns about practical applicability and foundational assumptions persist.

---

### Decision · Program_Chairs · 2026-01-26

Reject